# Feasibility of whole genome and transcriptome profiling in pediatric and young adult cancers

N. Shukla[1,8], M. F. Levine [1,2,8], G. Gundem [1,2,8], D. Domenico[1,2], B. Spitzer[1], N. Bouvier[1], J. E. Arango-Ossa[1,2], D. Glodzik[2], J. S. Medina-Martínez[2], U. Bhanot [3,4], J. Gutiérrez-Abril[1,2], Y. Zhou[2], E. Fiala[1,5], E. Stockfisch[1], S. Li[1], M. I. Rodriguez-Sanchez[1], T. O'Donohue[1], C. Cobbs [6], M. H. A. Roehrl [3,4,7], J. Benhamida [3], F. Iglesias Cardenas[1], M. Ortiz[1], M. Kinnaman[1], S. Roberts[1], M. Ladanyi[3], S. Modak[1], S. Farouk-Sait[3], E. Slotkin[1], M. A. Karajannis [1], F. Dela Cruz[1], J. Glade Bender [1], A. Zehir [3], A. Viale[6], M. F. Walsh [1,5], A. L. Kung [1,9✉] & E. Papaemmanuil [1,2,9✉]

The utility of cancer whole genome and transcriptome sequencing (cWGTS) in oncology is increasingly recognized. However, implementation of cWGTS is challenged by the need to deliver results within clinically relevant timeframes, concerns about assay sensitivity, reporting and prioritization of findings. In a prospective research study we develop a workflow that reports comprehensive cWGTS results in 9 days. Comparison of cWGTS to diagnostic panel assays demonstrates the potential of cWGTS to capture all clinically reported mutations with comparable sensitivity in a single workflow. Benchmarking identifies a minimum of 80× as optimal depth for clinical WGS sequencing. Integration of germline, somatic DNA and RNA-seq data enable data-driven variant prioritization and reporting, with oncogenic findings reported in 54% more patients than standard of care. These results establish key technical considerations for the implementation of cWGTS as an integrated test in clinical oncology.

[1] Department of Pediatrics, Memorial Sloan Kettering Cancer Center, New York, NY, USA. [2] Department of Epidemiology & Biostatistics, Memorial Sloan Kettering Cancer Center, New York, NY, USA. [3] Department of Pathology, Memorial Sloan Kettering Cancer Center, New York, NY, USA. [4] Precision Pathology Biobanking Center, Memorial Sloan Kettering Cancer Center, New York, NY, USA. [5] Department of Medicine, Memorial Sloan Kettering Cancer Center, New York, NY, USA. [6] Integrated Genomics Operation Core, Center for Molecular Oncology, Memorial Sloan Kettering Cancer Center, New York, NY, USA. [7] Human Oncology and Pathogenesis Program, Memorial Sloan Kettering Cancer Center, New York, NY, USA. [8] These authors contributed equally: Shukla N, Levine MF, Gundem G. [9] These authors jointly supervised this work: Kung AL, Papaemmanuil E. ✉email: kunga@mskcc.org; papaemme@mskcc.org

Cancer is caused by the accumulation of somatic variants, including point mutations, structural variants (SVs), and copy number alterations (CNAs) that drive oncogenesis, disease progression, and in some cases define therapeutic vulnerabilities. The introduction of next-generation sequencing (NGS)-based targeted gene-panel assays has aided disease diagnosis, guided care, and improved patient outcomes through refinement of treatment options[1–5]. However, targeted panels are optimized to assess clinical biomarkers in common cancers[1,2,4]. Recent studies showed the utility of whole-exome sequencing (WES) in identifying coding mutations in rare cancer genes[6,7]. However, for patients with pediatric or rare cancers that have low mutation burden, distinctive methylation profiles[8,9] are primarily driven by SVs and fusion genes, panel/WES tests fail to identify a clinical biomarker in most cases[1–4,7]. This underscores an unmet need for better diagnostic workflows to guide clinical management.

Cancer whole-genome and transcriptome sequencing (cWGTS) offers the opportunity to assess the full spectrum of germline and somatically acquired mutations, SVs and CNAs, along with quantification of tumor mutation burden (TMB) and genome-wide mutational patterns[10]. The likely clinical utility of cWGTS in pediatric and rare cancers is increasingly evidenced in recent literature[7,11–16]. However, clinical implementation of WGS in oncology is challenged by cost of sequencing, complexity of laboratory, and analytical workflows to process large-scale data within clinically relevant timeframes, concerns about the sensitivity of low-coverage WGS in detecting actionable mutations captured by high-depth panel assays, and the interpretability of cWGTS findings with regard to clinical utility[3]. Here, we demonstrate the feasibility, analytical validity, and resolve critical technical considerations for the implementation of cWGTS in primary cancer care in the context of pediatric, adolescent, and young adult solid tumor patients with rare cancers.

## Results

**Sample processing**. The study cohort included patients presenting with primary diagnostic or relapse/refractory disease. Of 201 patient fresh frozen (FF) tumors nominated for paired cancer/normal whole-genome and transcriptome sequencing (cWGTS), 58 were excluded upon pathology review and 29 did not meet our requirement for >20% tumor purity as assessed by WGS. The majority of the excluded cases were post-therapy neuroblastoma and sarcoma samples with a predominance of necrotic disease. The final cohort included a single sample each from 114 pediatric, adolescent, and young adult patients (median age = 12.6 years, range: 4.5 months to 43.8 years) with solid tumors (Supplementary Tables 1, 2, Supplementary Fig. 1a–d).

**Implementation of a cWGTS workflow for clinical decision support**. To prototype a clinical cWGTS workflow, we developed an end-to-end process (Fig. 1a) that included dedicated: 1. project-management team, 2. lab operators for sample processing, 3. sequencing machines for cWGTS, 4. data-import channel, 5. Biosciences platform for automated deployment of analysis pipelines and API integration with institutional and public databases[17], 6. reserved computing nodes in a high-performance computing environment, and 7. systematic pipeline for prioritization and reporting of genomic findings (Fig. 1a, Supplementary Fig. 1e). We quantified the end-to-end time from sample acquisition to the generation of an automated report. Time logs were audited starting at the time of surgical biopsy submission to report delivery for review by an interdisciplinary molecular tumor board for samples in our study with audit trails recorded through our biosciences platform (Supplementary Fig. 1e). End-to-end,

this workflow was executed on average in 17 days during the developmental phase (range: 11–29, $n = 59$ samples), reaching a fully optimized workflow with a final turnaround of 9 days (Fig. 1b, $n = 16$). This is shorter than the standard turnaround time (TAT) for many clinical NGS-panel sequencing tests (2–4 weeks)[1,2,4] markedly faster than the majority of WGS-processing timeframes in literature (3–8 weeks, Fig. 1b)[11–16] and comparable to the TAT achieved by centralized infrastructures of scale such as the Hartwig Foundation[18]. This demonstrates the feasibility of implementing cWGTS profiling to support diagnosis and treatment decisions with a clinically relevant turnaround time within a comprehensive cancer care center.

**Comprehensive genome characterization utilizing cWGTS**. Across all mutational classes, cWGTS identified on average 7353 acquired mutations per sample, including cancer-associated alterations in 99% ($n = 114$) of patients (Supplementary Fig. 2). These include CNAs ($n = 105$ patients), germline predisposition ($n = 17$), mutations in cancer-associated genes ($n = 77$), translocations/fusion transcripts ($n = 27$), disease-associated SVs ($n = 75$), and outlier TMB or microsatellite-instability (MSI) scores ($n = 7$) (Supplementary Table 3). Further signals of interest included the delineation of mutation signatures[19], detection of chromothripsis[20] or whole-genome duplication (WGD)[21], cancer-associated viral sequences (i.e., EBV)[22,23], estimation of telomere length[24], and gene expression signatures. SVs, most of which can only be detected using WGS, represented the third most frequent class of genomic alterations.

**Concordance analysis of cWGTS to targeted DNA- and RNA-panel tests**. Within our cohort, targeted DNA profiling of corresponding formalin-fixed paraffin-embedded (FFPE) biopsies by MSK-IMPACT[4] detected actionable biomarkers as defined by OncoKb Levels 1–4[25] in 24% of patients ($n = 27$) (Fig. 2a–c, Supplementary Table 4). Consistent with prior findings demonstrating that patients with rare cancers do not yield clinically relevant biomarkers by panel sequencing[4], most patients in our cohort (76%, $n = 87$) had no therapy-informing alterations. These results are representative of the expanded pediatric/young-adult patient population at MSK (Supplementary Fig. 3a, b).

We first assessed whether mutations captured by MSK-IMPACT were also detected by WGS. For all discordant samples, we performed MSK-IMPACT on the same DNA aliquots used to generate the WGS libraries. This allowed us to ascertain whether discrepant calls were owing to differences in assay sensitivity (MSK-IMPACT and WGS) or a consequence of intratumor heterogeneity (ITH)[26].

Of 221 somatic mutations reported by MSK-IMPACT, 174 (79%) were called in WGS (Fig. 2d, e). This includes 68/83 (82%) mutations reported by MSK-IMPACT as oncogenic[25] (Supplementary Fig. 3c). Variants called by both assays ranged from 5% to 97% variant allele frequency (VAF) with high concordance ($r^2 = 0.75$) in VAF estimates (Fig. 2d). The majority of discordant mutations (46/47) were subclonal in MSK-IMPACT (<90% of cancer cell fraction) and 15 were classified as oncogenic (Supplementary Table 4, Supplementary Fig. 3c). Discordant mutations presented with a broad range of VAF (range: 2.2–39%, median = 8.5%) (Fig. 2d) and showed no systematic bias in effective coverage (Supplementary Fig. 3d). The 47 discordant mutations were confined to 26 samples (range 1–7 mutations per patient). Targeted resequencing of the WGS libraries by MSK-IMPACT was performed for 44 discordant variants and none were called, despite a median local depth of sequencing at 469x, supporting ITH as the basis of the discrepancies (Supplementary Fig. 3e). Further corroborating ITH, WGS and targeted

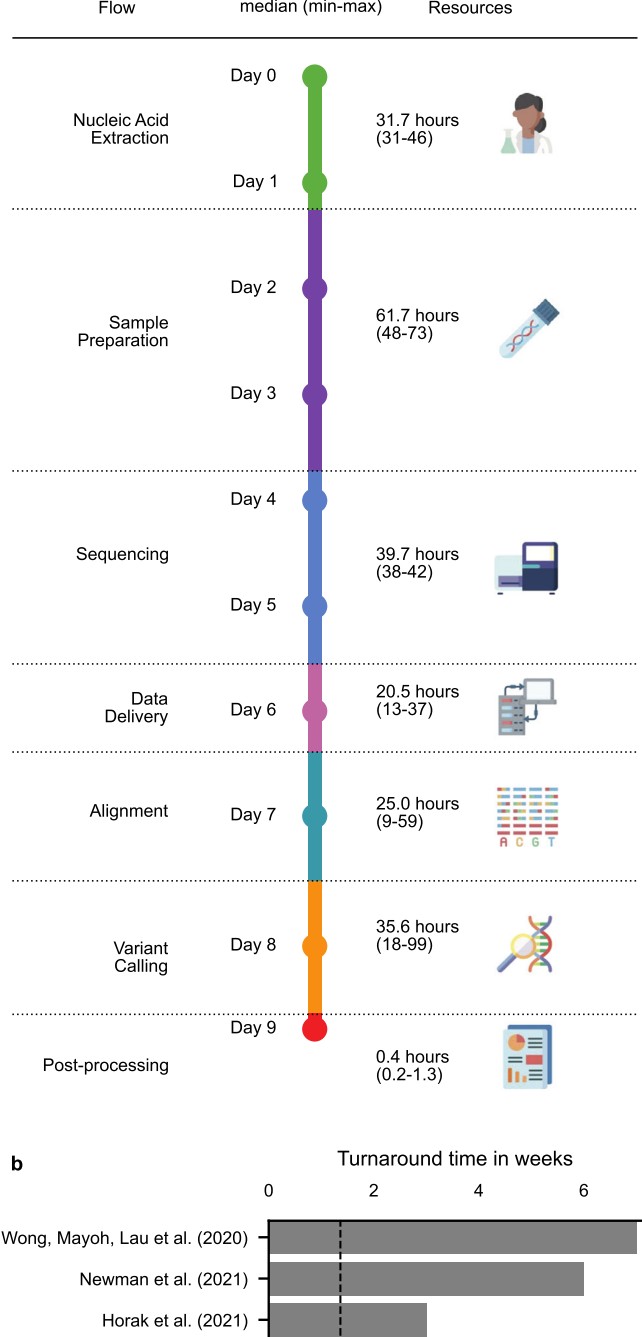

**Fig. 1 End-to-end cWGTS workflow. a** Schematic representation of the end-to-end cWGTS workflow, with information on median-time duration (in hours) for each step, as determined by a time trial over four consecutive batches containing $n = 16$ tumors and representation of dedicated resources necessary to execute the workflow. **b** Comparison of best-reported turnaround times in literature, from sample collection to results ready for tumor board review. For our study, we show an orange bar denoting median time for $n = 16$ samples with minimum and maximum times denoted with the error bar. These samples were processed post optimization.

resequencing data of the same FF DNA aliquots identified 10 mutations (VAF: 6–31%), which were not reported by MSK-IMPACT from the patient-matched FFPE sample. Three of the 10 additional calls were cancer-associated variants (TP53 L265Yfs*81, PPM1D S468*, and HLA-A L102Hfs*73) (Supplementary Fig. 3e).

These validation studies demonstrated that discordant calls were due to stochastic sampling of heterogeneous tumors[26] (Supplementary Table 4) and concordance between WGS and MSK-IMPACT is at least 94% (43/47 total discordant) and up to 100% for all 43 mutations in evaluable samples when the same DNA aliquot was used in both assays.

Germline assessment by MSK-IMPACT[27] identified predisposition variants in 13 patients and panel RNA assessment with MSK-Fusion[28] identified oncogenic fusion genes in 18 (Supplementary Tables 5, 6, Fig. 2f). cWGTS captured all 13 germline-predisposition variants and 18 fusions. Importantly, fusion genes were supported by data in both WGS and RNA-seq, which offers the opportunity to orthogonally validate findings within a single workflow (Fig. 2f). These findings demonstrate that cWGTS as an integrative assay allows for the detection of germline, somatic mutations and fusion genes captured by an array of standard-of-care diagnostic tests.

**Technical considerations: optimal depth of coverage for clinical sequencing**. Sensitivity for somatic variant detection is directly dependent on the tumor cellularity of the biopsy and depth of sequencing coverage. The median cWGS depth was 95x (range 67–181) and tumor purity ranged from 21% to 100%, resulting in a median effective coverage of 64× (median depth * purity estimate). To evaluate optimal depth of sequencing, for each of 97 tumors with WGS coverage ≥60×, we generated 298 derivative subsampled BAM files in the range of 100×, 80×, 60×, and 30–40× (Supplementary Fig. 4a, Supplementary Table 7). De novo variant calling was performed to assess sensitivity of detection for clinically relevant findings by MSK-IMPACT and WGS ($n = 220$), genome-wide mutations across variant classes, and TMB (Fig. 3a, Supplementary Fig. 4b, c). Detection sensitivity correlated with effective coverage and was affected by variant class with slightly less sensitivity for SVs (Fig. 3a). Of the oncogenic findings, >91% were captured at 30–40× and >98% were recalled at 60–100× (Fig. 3a). With lower sequencing coverage, the power to detect subclones is limited (VAF range: 4.3–31%, median = 9.5%) (Fig. 3b, c). Optimal sensitivity for genome-wide mutation calling across variant classes was attained at ≥80× and increased with coverage. Figure 3c provides an overview of variant-detection sensitivity by depth of sequencing coverage, tumor purity, and variant clonal representation.

**Findings of biological and clinical relevance detected by cWGTS only**. The clinical relevance of cWGTS findings that were not identified by clinical panel sequencing (MSK-IMPACT[4], MSK-fusion[28] and panel testing of 88 cancer-predisposition genes[27]) was determined by a multidisciplinary molecular tumor board. Consistent with recent studies[7,11–16], cWGTS analyses identified at least one additional cancer-associated oncogenic variant in 54% of patients ($n = 62$). Of these, 33 patients had one or more findings that were of direct clinical relevance, including 7 diagnostic (21%), 15 prognostic (45%), 5 therapy-informing (15%), 5 previously undescribed oncofusions (15%), and 6 germline (18%) biomarkers (Fig. 4a, Supplementary Table 3). Most additional relevant findings were explained by the detection of SVs and fusions and genome-wide mutation signatures (Fig. 4b).

We further inferred the portion of additional findings that would be captured by WES by masking results to the coding regions of genes. Of the 62 patients with incremental findings by

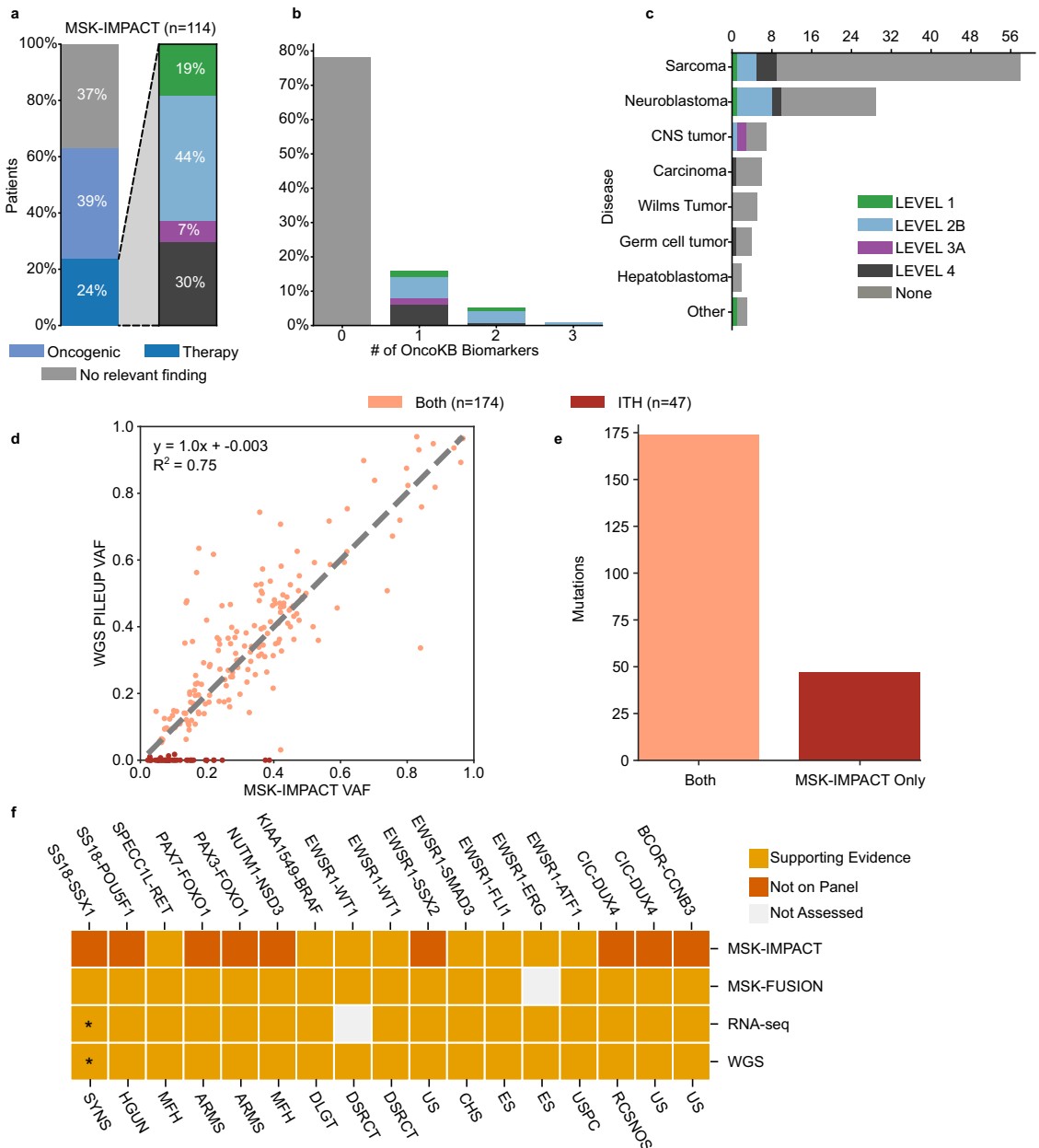

**Fig. 2 Analytical validity of cWGTS for clinical biomarkers. a** The left barplot depicts the proportion of patients with therapy-informing, oncogenic, or no relevant findings reported by MSK-IMPACT as defined by OncoKb (Levels 1–4). The right barplot shows the breakdown (0,1,2) of the highest level of OncoKb annotation in the study cohort. **b** Barplot demonstrating breakdown of the highest OncoKb level by the number of informative biomarkers in study cohort. **c** Barplot demonstrating breakdown of the highest OncoKb level by disease class. **d** Scatterplot shows the comparison of variant allele frequency (VAF) of MSK-IMPACT variants as reported by MSK-IMPACT (x axis) and absolute VAF estimates by pileup in WGS data (y axis) (Pearson correlation). Discrepant mutations are observed along the x axis. Mutations are color-coded by call status, where Both is called in both assays and ITH is mutations that were not called in higher- depth resequencing and/or had proportion test *p*-value < 0.05. **e** Barplot demonstrating breakdown of MSK-IMPACT mutations, observed in both WGS and MSK-IMPACT or only MSK-IMPACT (ITH). **f** Validation of oncogenic fusions reported by MSK-IMPACT/MSK-Fusion in cWGTS. The asterisk indicates that the *SS18-SSX1* that was reported by MSK-Fusion was reported as *SS18-SSX2* by RNA-seq and supported by spanning reads in WGS. Main oncotree disease code listed underneath for each patient (ARMS alveolar rhabdomyosarcoma, CHS chondrosarcoma, DLGT diffuse leptomeningeal glioneural tumor, DSRCT desmoplastic small round-cell tumor, ES Ewing sarcoma, MBL medulloblastoma, MFH undifferentiated pleomorphic sarcoma/malignant fibrous histiocytoma/high-grade spindle-cell sarcoma, RCSNOS round-cell sarcoma, NOS, SYNS synovial sarcoma, US undifferentiated sarcoma, USPC undifferentiated sarcoma of the peritoneal cavity). Source data for panels a–e and f are provided in Supplementary Data 4 and 6.

cWGTS, RNA-seq and WES alone would only capture events in 10 (16%) and 8 (*n* = 13%) patients, respectively, or in 17 patients when combined (Fig. 4a, Supplementary Table 3). Thus, only 27% of the findings in cWGTS could be captured by WES and RNA-seq as the majority of additional findings were attributed to SVs.

**Rare variants in established cancer genes**. We identified seven clinically relevant findings targeting known rare cancer genes (Supplementary Tables 5, 8). Of these, three were somatically acquired and included a disease-defining mutation of *KBTBD4* (p.R313_M314insPRR) in a pineal parenchymal tumor of

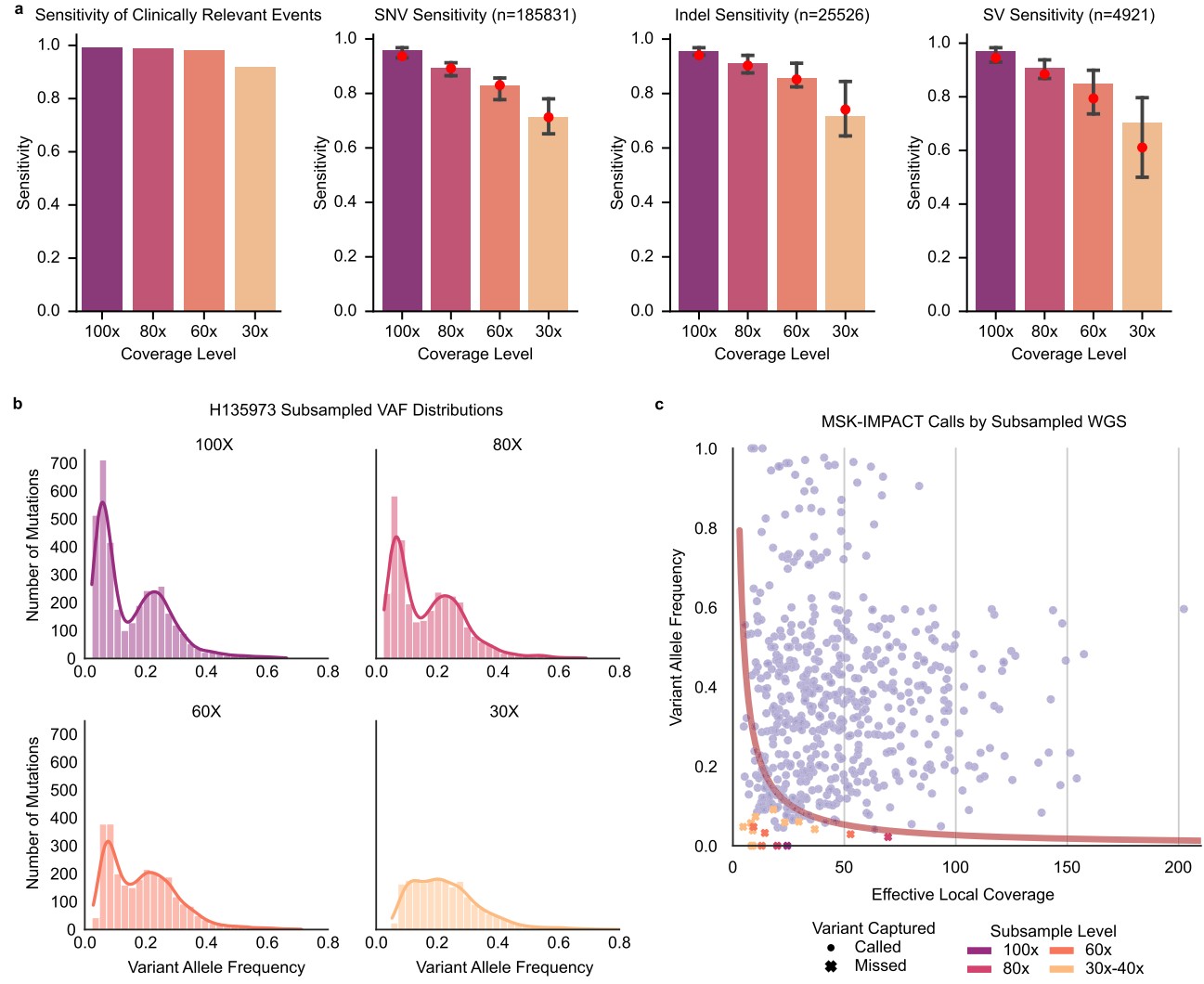

**Fig. 3 Assessment of optimal coverage for WGS. a** Barplots demonstrating sensitivity of variant detection and 95% confidence intervals (error bars) by coverage depth (100x, 80x, 60x, and 30–40x) from left to right for: 1. clinically relevant events detected by MSK-IMPACT and WGS ($n = 220$), 2. genome-wide SNVs, 3. genome-wide indels, and 4. genome-wide SVs. Only data from samples with original median coverage >100x ($n = 32$) are shown. Red dots indicate overall sensitivity of all mutations across all BAMs at the same subsampling level. **b** Histograms of variant allele frequencies for each subsampling level for a representative sample in the study cohort (H135973), showing loss in sensitivity to detect subclonal mutations at lower sequencing depth of coverage. **c** Scatterplot of effective local coverage vs VAF in subsampled BAMs for the clinically relevant calls from MSK-IMPACT. Variants called in subsampled BAMs are shown with circles, while the missed variants are denoted with X's. Trendline shows the cumulative binomial distribution for obtaining at least 2 variant reads, given the effective coverage and variant allele fraction. Source data for panels **a**, **c** are provided at the data repository. Raw data for panel **b** can be accessed at the dbGAP study.

intermediate differentiation[29,30], a *SETBP1* (p.D868N) mutation in a germ cell tumor, and a *SIX1* mutation (p.Q177R) in a Wilms tumor[31]. Additionally, clinically relevant germline variants were detected in four cancer-associated genes, including an *SBDS* splice-site mutation (c.258 + 2T > C) in a rhabdomyosarcoma, *BARD1* p.E652fs*69 in a neuroblastoma, *EP300* p.A2259fs*20, and *EXT2* p.W414* in two osteosarcoma patients (Supplementary Table 5). These results demonstrate the utility of cWGTS in capturing somatic and germline variants in rare cancer genes not routinely evaluated in targeted panels[2,4].

**Fusion genes.** Eight in-frame fusion genes were identified from WGS and RNA-seq in patients with no prior findings on clinical testing (Supplementary Fig. 5, Supplementary Table 6), 5 of which were not described before. Of diagnostic relevance, we identify: 1. a t(2;6) (*PAX3-FOXO3*) translocation changing

diagnosis to alveolar rhabdomyosarcoma (ARMS) in a patient who was diagnosed with embryonal rhabdomyosarcoma (ERMS) in the absence of the cardinal ARMS fusions (*PAX3-FOXO1* and *PAX7-FOXO1*)[32] (Fig. 5a, b), 2. a *UACA-LTK* fusion in a metastatic papillary thyroid carcinoma[33], and 3. a pathogno-monic *SH3PXD2A-HTRA1* fusion establishing a diagnosis of schwannoma in a patient evaluated for relapsed stage-IV neuroblastoma[34].

Of potential therapeutic relevance, we identified an *NTRK3-SLMAP* fusion in a neuroblastoma patient. Activating *NTRK3* fusions are promising therapeutic targets for TRK inhibitors, with activity seen across pediatric and adult cancers[35]. However, screening for *NTRK* fusions is not routinely performed across all disease indications. Additional undescribed fusions included *EPC2-AFF3* and *MAN1A2-ACBD6* identified in two patients with undifferentiated sarcoma, and a *CITED2-MGA* fusion in a round-cell sarcoma not otherwise specified.

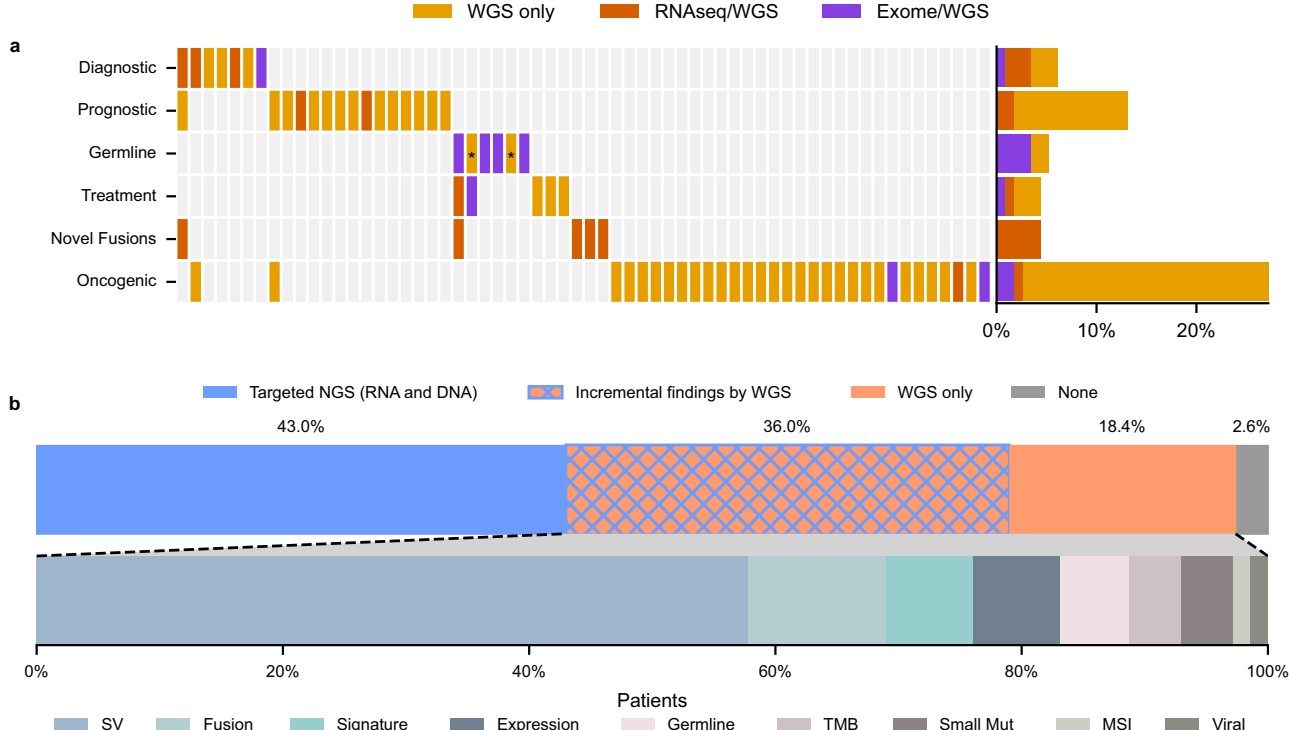

**Fig. 4 Additional relevant findings detected by cWGTS as compared with standard of care. a** Heatmap of additional relevant findings by cWGTS colored by what technology (WES, WGS, and RNA-seq) may detect each event. Columns represent patients, while rows are clinical event types. The asterisks for Germline indicate pathogenicity supported by mutational signatures. **b** (top) Stacked-bar breakdown of patients with clinically relevant findings by assay. The blue areas (solid or meshed) represent patients with relevant findings from targeted sequencing (RNA and DNA), while the orange areas (solid or meshed) are for patients with findings from cWGTS. The blue/orange mesh indicates patients that had relevant findings from both targeted sequencing and WGTS. (bottom) Stacked-bar breakdown of findings specific to cWGTS from the patients in the orange section (solid or meshed) from top. The relevant findings are colored by event type. SV, structural variant. TMB, tumor mutation burden. MSI, microsatellite instability. Small Mut, small mutations, including substitutions and insertion/deletions. Viral, viral integration. Source data for panels **a**, **b** are provided in Supplementary Data 3.

**Structural variants targeting tumor suppressor genes**. Structural variations of established prognostic relevance[36] were observed in our cohort providing insights of clinical relevance. cWGTS mapped events in *TERT* and *ATRX* in 8 (28%) and 5 (17%) neuroblastoma patients, respectively. Both *TERT* and *ATRX* are increasingly considered as therapy-defining risk-stratification biomarkers for neuroblastoma[37]. The *TERT* SVs could only be identified by cWGTS, and only ¼ of the *ATRX* deletions were reported by MSK-IMPACT[4].

We also observed recurrent SVs targeting the tumor suppressor gene *DLG2* in 15/29 OS patients[38] and 3/29 neuroblastoma, of which 6 had homozygous deletions (Supplementary Table 9). While *DLG2* has been characterized in osteosarcoma[38], our findings demonstrate that *DLG2* SVs are also recurrent in neuroblastoma warranting further investigation in future studies.

**Integration of RNA-seq and WGS for variant annotation**. Interpretation of complex SVs in noncoding regions of the genome presents a major challenge for reporting of WGS findings. cWGTS enables the concomitant detection of SVs and assessment of the transcriptomic consequences of the affected loci. For example, a chromoplexy event resulting in overexpression of the *MYB* oncogene[39] through "hijacking" of an *NFIB* enhancer (Fig. 5c, d) was detected in an adenoid cystic carcinoma without informative clinical sequencing findings. While *MYB* overexpression is a cardinal feature of adenoid cystic carcinomas, *MYB* fusions are identified in only 30% of cases using conventional diagnostic assays[39]. Integration of gene expression data was

critical to the annotation and reporting of this complex noncoding SV as the disease defining diagnostic biomarker.

Similarly, among 29 osteosarcoma patients, we identified *TP53* mutations in 12 and mapped noncoding SVs targeting the *TP53* locus in 13. Of these, only 3 were reported by MSK-IMPACT (Fig. 5e). Integration with RNA-seq demonstrated that *TP53* SVs correlated with loss of *TP53* expression, validating their functional relevance (Fig. 5f). Wild-type *TP53* represents an inclusion criterion for p53 pathway modulating drug trials[40]. Here, we show that in the absence of cWGTS, patients with loss of *TP53* by SVs, which have been described in diverse cancers, could be erroneously diagnosed as *TP53* wildtype with implications for assessment of treatment options[40]. We did not identify germline SVs targeting the *TP53* locus.

Taken together, our findings illustrate the necessity to combine RNA and DNA analyses in variant detection, annotation, and prioritization for clinical cWGTS reporting. In our automated workflow, we interrogate DNA mutations for corroborating evidence in the RNA. All recurrent fusion genes reported by panel RNA-seq assays as well as the 8 additional driver fusion genes were orthogonally detected by both WGS and RNA-seq (Supplementary Fig. 5). Furthermore, integration of gene expression data to SV findings resolved functional consequences of SVs targeting noncoding regions on the genome (e.g., *MYB* enhancer hijacking and *TP53* inactivation).

Global gene expression signatures were further used to cluster samples by tumor type, providing further opportunity to resolve a patient's diagnosis (Supplementary Fig. 6a). Last, in the 101 patients with RNA-seq data, we identified on average 18 gene expression

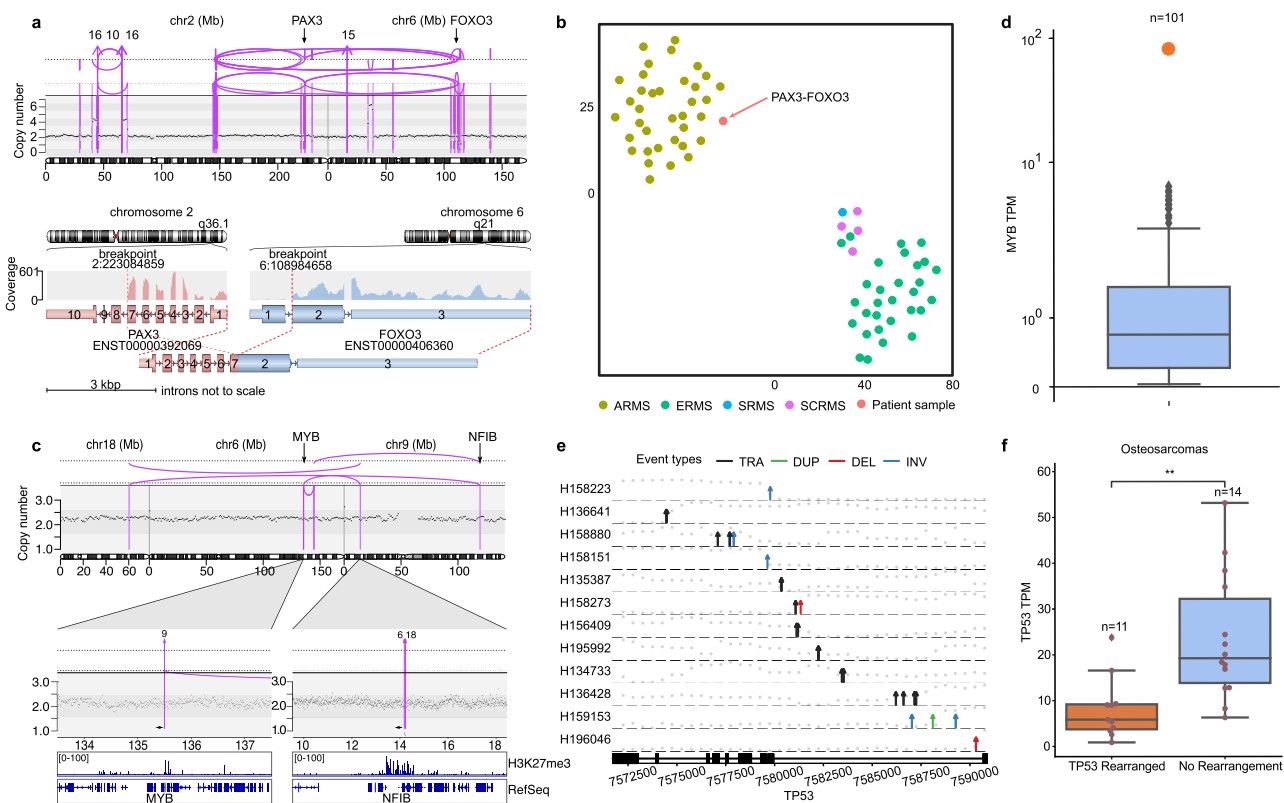

**Fig. 5 Integration of DNA and RNA findings for variant annotation. a** Top panel shows absolute copy number on the y axis and the structural variants (SVs) that result in *PAX3-FOXO3* fusion in patient H134768. Lower panel displays RNA fusion product created by the corresponding genomic SVs. **b** tSNE clustering of methylation data from rhabdomyosarcoma samples color-coded by disease subtype (ARMS: alveolar, ERMS: embryonal, SCRMS: spindle cell, and SRMS: sclerosing). The patient harboring the *PAX3-FOXO3* fusion clusters with the ARMS samples. **c** Top panel shows the chromoplexy event among chromosomes 6, 9, and 18, resulting in the localization of *NFIB* enhancer to the *MYB* locus in patient H133676. Lower panel displays H3K27me3 chromatin marks from Drier et al., Nature Genetics 2016. **d** Boxplot shows the *MYB* expression in transcripts per million (TPM) across the cohort. Center line indicates the median and whiskers extend within $+/-1.5$x the interquartile range (IQR) from the box. The patient with *MYB-NFIB* event (H133676) is highlighted in orange, demonstrating that the SV event in panels **c** associates with overexpression of MYB, validating the SV as an enhancer-hijacking event. **e** Diagram of SV events targeting *TP53* gene body in osteosarcoma patients ($n = 12$, the 13th patient's event breakpoints fall outside of the gene body). SVs are shown as arrows with absolute copy number on the y axis (gray dots) overlaid over the exonic structure of *TP53* (TRA: translocation, DUP: duplication, DEL: deletion, INV: inversion). **f** Boxplot shows the comparison of *TP53* expression in RNA between *TP53*-rearranged samples and those without any rearrangement with a center line indicating the median and whiskers extending within $+/-1.5$ x the IQR (two-sided Mann–Whitney U test, $p = 1.645$e-03). Raw data for panel **a**–**c** can be accessed at the dbGAP study. Source data for panel **e** are provided in Supplementary Data 9. Source data for panels **d**, **f** are provided at the data repository.

biomarkers per sample (range = 1–99)[16] (Supplementary Table 10). However, the evidence with regard to the clinical utility of such expression biomarkers remains to be validated. To this end, we performed a systematic interrogation of genes with aberrant expression with known tissue-specific expression patterns and SVs or fusions detected from cWGTS. Overall, only 8% of the expression biomarkers were associated with an acquired SV or fusion gene, demarcating a subset of high-confidence expression biomarkers (average per patient = 1, range = 0–10). This limits the number of patients in our cohort with an expression biomarker supported by an SV to 54% ($n = 55$) (Supplementary Fig. 6b). Demonstrating the utility of this integrative analysis, we identified a concomitant *KRAS* amplification and overexpression in two patients with no clinically relevant biomarkers (H135462 and H195916) by clinical testing. Of note, in both patients, the amplification was not reported by MSK-IMPACT pointing to the lower sensitivity to detect copy number changes by panel-based assays.

**Integration of germline mutations to somatic mutation signatures for variant annotation**. Annotation of germline variants is restricted to recurrent events in population databases, thus

limiting interpretation for rare founder events. For each genome in our cohort, we quantified the proportion of mutations attributed to each of 73 reference mutation signatures[10]. Two of three patients with a germline mutation in DNA repair genes further harbored mutation profiles suggestive of DNA repair deficiency. Patient H211942 had a pathogenic variant in *MUTYH* and somatic loss of the second allele. About 42% of the mutations were attributed to the *MUTYH* signature SBS36[10] (Fig. 6a). Patient H135466 had a pathogenic variant in *PMS2* (c.538–1G > C) with loss of the wild-type allele by LOH. The tumor was MSI high with hypermutation (TMB = 11.23, indels = 90,246, SNVs = 17,840, and SVs = 44), enrichment of T > C mutations, and repeat-mediated indels characteristic of *PMS2* deficiency[10] (Fig. 6b). In contrast, patient H135073 harbored a variant of unknown significance (VUS) in *PMS2*, a medium MSI score (7.23), and low mutation burden (1.30 Muts/Mb) without evidence of a *PMS2* signature (Fig. 6c). These findings demonstrate the utility of mutation signatures in the assessment of germline mutations in DNA repair genes.

To illustrate this point, in a 12-year-old osteosarcoma patient outside the study cohort, cWGTS characterized a hypermutated genome (TMB = 16.7, indel = 89,588, SNVs = 31,520, and

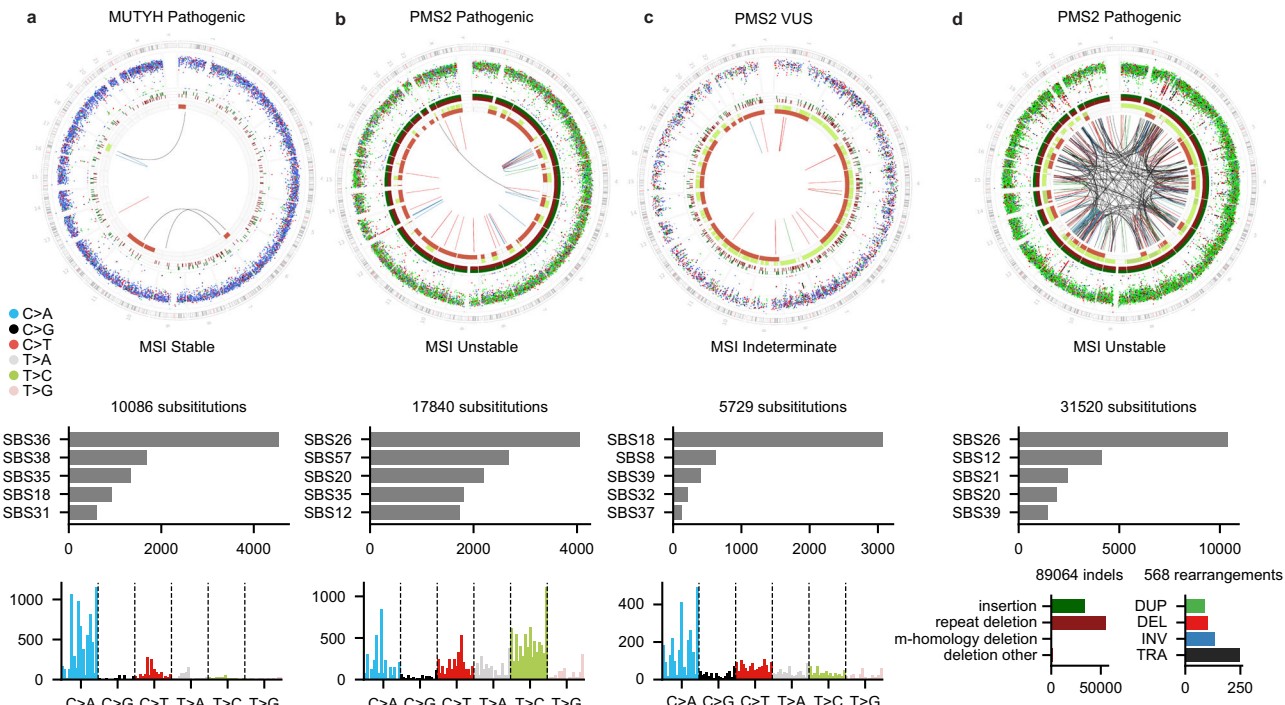

**Fig. 6 Genome-wide distribution and patterns of somatic mutations for four different patients. a** Neuroblastoma patient (H135421) harboring a pathogenic germline *MUTYH* variant (c.924 + 3A > C). **b** Immature teratoma patient (H135466) with a pathogenic germline *PMS2* mutation (c.538-1G > C). **c** Malignant peripheral nerve sheath tumor patient (H135073) harboring a germline *PMS2* variant of unknown significance (VUS) (p.W841*). For each patient, the top panel is a Circos plot showing the different types of somatic mutations along the genome. The outermost ring shows the intermutation distance for all SNVs color-coded by the pyrimidine partner of the mutated base. The middle ring shows small insertions (green) and deletions (red). The innermost ring shows copy number changes, and the arcs show SVs. Middle panel is a barplot showing the absolute number of mutations attributed to the five mutational signatures with the highest exposure in the tumor. Bottom panel is a barplot showing the 96 trinucleotide contexts of SNVs. **d** Genome-wide distribution and patterns of somatic mutations identified in the patient outside the cohort with recurrent osteosarcoma (H201472). WGS results show the sample is hypermutated, with enrichment in SBS26, T > C mutations, repeat-mediated deletions, and MSI unstable. The patient was found to be harboring a pathogenic *PMS2* variant (p.D699H) (repeat deletion: repeat-mediated deletion, m-homology: microhomology-mediated deletion, deletion other: all other deletions, TRA translocation, DUP duplication, DEL deletion, INV inversion). Raw data for this figure can be accessed at the dbGAP study.

SVs = 568) enriched in repeat-mediated deletions consistent with MSI high status (Fig. 6d). This observation prompted consent for germline testing, resulting in identification of a *PMS2* mutation (p.D699H) annotated as likely pathogenic/VUS[41] and a somatic loss of the wild-type allele. MSK-IMPACT reported an indeterminate MSI status (7.5) yet upon testing validated that the germline mutation is pathogenic.

These results demonstrate the utility of integrating composite readouts from cWGTS (germline mutation, allele-specific copy number, and genome-wide TMB) to deliver corroborating evidence for the assessment and reporting of germline-predisposition mutations with implications for family screening, diagnosis, and treatment.

### Genomic alterations of emerging biological and clinical relevance.
Recent studies propose telomere length as prognostic indicators in neuroblastoma among other cancers[42–45]. We recapitulated the established associations between *ATRX* and *TERT* mutations to telomere length[46,47] (Supplementary Fig. 7a–c). *ATRX* mutations were also observed in 8 osteosarcomas, with similar associations to telomere length (Supplementary Fig. 7c). Given the association between adverse risk mutations and telomere length, delineation of the independent prognostic value warrants analyses of data that concomitantly map these mutations, SVs, and telomere length alongside established predictors of outcomes.

We detect chromothripsis[20] in 40% of patients, most recurrently observed in sarcomas (35/58), germ cell tumors (2/4), and less frequently in neuroblastoma (6/29) (Supplementary Fig. 7d). Chromothripsis frequently led to *TP53* loss (10/29)[20,48], amplification of *MYC*, *VEGFA*, and *MDM2*. Additionally, in 2 patients, chromothripsis resulted in oncogenic fusions (*MAN1A2-ACBD6* and *PAX3-FOXO3*) (Supplementary Fig. 7e). Previous studies have proposed an association between whole-genome duplication (WGD) and poor outcomes in cancer[21]. WGD was seen in 42/114 patients with an enrichment in sarcoma (24/54), carcinomas (3/7), and neuroblastoma (11/30) (Supplementary Fig. 7f).

### Biological and clinical implications of tumor mutation burden across variant classes.
Panel-based approaches derive estimates of TMB, MSI scores, and mutation signatures[49], whereas WGS directly quantifies genome-wide mutation burden across all variant classes (Fig. 7a, b and Supplementary Fig. 8a). We observed higher overall (8.3-fold) TMB estimates in our cohort, relative to reports in pediatric cancer (Fig. 7a)[10,48,50]. TMB was higher in therapy-exposed compared with treatment-naive patient samples (0.1–11.2 in treated vs 0–2.7 treatment-naive, Mann–Whitney test, $p = 1.892e-04$) and correlated with evidence of treatment-related signatures (i.e., temozolomide, platinum) (Supplementary Fig. 8b)[10,51], observed in 45/114 patients pointing to persistence of clones that were exposed to and survived cancer therapy[52].

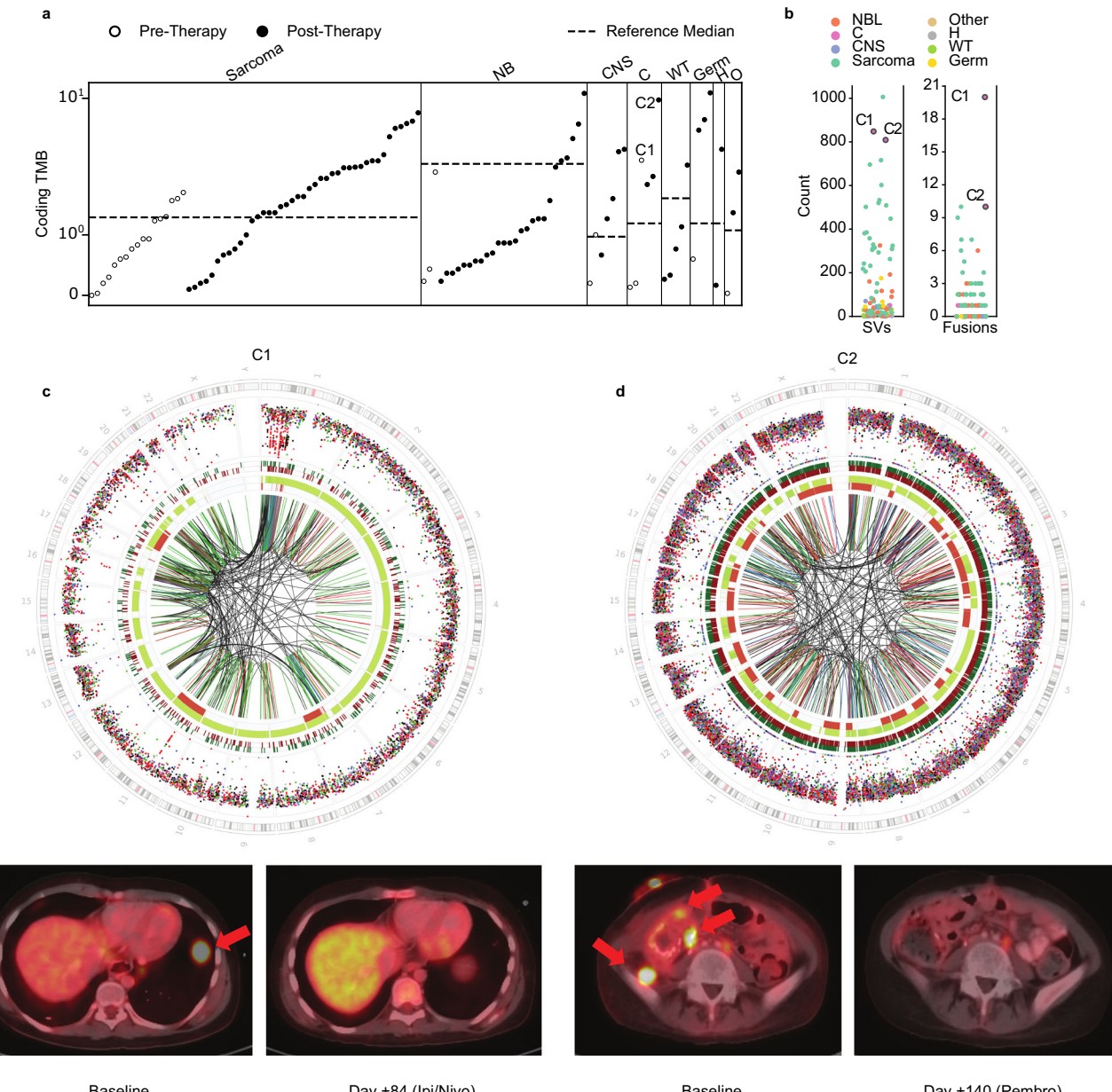

**Fig. 7 Genome-wide mutational burden in the context of immunotherapy. a** Distribution of coding tumor mutational burden (TMB) as assessed by WGS across the cohort (*n* = 114), colored by treatment status of the patient at the time of sampling. Dotted line indicates median-coding TMB (SNVs and indels) as previously reported by the Zero Childhood Cancer study. Patients are grouped by disease category (NB: neuroblastoma, CNS: central nervous system, C: carcinoma, WT: Wilms tumor, Germ: germ cell tumor, H: hepatoblastoma, O: other). Carcinoma patients C1 and C2 who responded to immunotherapy are labeled. **b** Distribution of structural variant (SV) (right) and gene fusion (left) burden across the samples with both WGS and RNA-seq available (*n* = 101). Patient C2 had a poor-quality RNA sample, so clonal fusions from another time point from the same patient are shown. **c** (top) Genome-wide distribution and patterns of somatic mutations for tumor C1 (H135022), patient with metastatic adrenocortical carcinoma, depicting high SV burden. Circos plots are shown as described in Fig. 6. PET imaging shows resolution of a large pulmonary metastatic lesion (red arrow) following treatment with nivolumab and ipilimumab. **d** Genome-wide distribution and patterns of somatic mutations for H135462, a 14-year-old with relapsed refractory poorly differentiated clear-cell carcinoma with high TMB and SV burden. Circos plots are shown as described in Fig. 5. PET imaging shows resolution of multiple metastatic lesions (red arrows) following treatment with pembrolizumab. Source data for panels a and b are provided at the data repository. Raw data for panel **c**, **d** can be accessed at the dbGAP study.

Patients H135022 (adrenocortical carcinoma) and H135462 (clear-cell carcinoma) had progressive on-treatment metastatic disease and in the absence of therapy-informing biomarkers by clinical testing were at the end of their therapeutic options. cWGTS analyses revealed a profoundly rearranged genome scoring these two patients as the highest in fusion burden and SV burden in the cohort (Fig. 7b–d). H135022 was treated with checkpoint blockade

(nivolumab/ipilimumab), resulting in complete response after three cycles of therapy, and is disease-free 26 months after therapy cessation (Fig. 7c), whereas patient H135462 was treated with pembrolizumab, achieved a complete response after 6 cycles, and remains disease-free 10 months after therapy (Fig. 7d).

These findings demonstrate the value of cWGTS to fully assess the level of genomic instability across variant classes and

highlight the need to further evaluate SV and fusion gene burden as biomarkers of response to immune checkpoint blockade therapies.

**Derivation of comprehensive WGS profiling in cell-free DNA**. Our study evaluated key technical considerations of cWGTS in FF biopsies, as an optimal source of tumor DNA. However, limited biopsies may restrict access to cWGTS for all patients. Cell-free DNA (cfDNA) from blood plasma represents an alternative source of DNA for tumor profiling[53]. Recently, cfDNA NGS profiling including WGS has been used to detect tumor-specific CNAs and fragmentation patterns in pediatric tumors[54–56]. However, the potential of high-depth WGS in tissue-naive identification of the genetic changes (SNVs, indels, and SVs) in a patient's cancer genome from cfDNA has been largely unexplored.

In an exploratory analysis, we performed WGS from matched FF samples and cfDNA from seven patients collected at the time of tumor biopsy (Supplementary Table 11). cfDNA genome-wide coverage ranged from 94 to 102× (Fig. 8a) with a wide range of tumor content in cfDNA ~10–83% (Fig. 8b). To assess the suitability of cfDNA for unbiased genome-wide mutation detection, we performed de novo mutation calling in cfDNA for all variant classes (SNVs, indels, SVs, and CNA) and compared the results to FF analyses.

WGS from cfDNA did not present technical limitations in data generation or false-positive variant calling across mutation classes. Derivation of high-quality variant calls was contingent upon the quantity of circulating tumor DNA (ctDNA). In four patients with ctDNA content sufficient for CNA detection, we establish good concordance between the FF and cfDNA CNA profiles (Fig. 8c, Supplementary Fig. 9a–g). Strikingly, for patients with high ctDNA content (i.e., IH158182), we derived a near-complete picture of the genome-wide mutation patterns demonstrating that cancer genomic landscape can be fully recapitulated by cfDNA WGS (Fig. 8d). Importantly, for patient H135967, we showcase that even with an estimated ctDNA content of 20%, the same threshold used for analyses of FF material, we can detect all the known oncogenic events in the FF sample across variant classes, which include a *TP53* substitution, *MYC* and *CCNE1* amplifications, and SVs targeting ARID1A and ATRX (Supplementary Fig. 9c).

We further demonstrate the potential of cfDNA to capture a comprehensive representation of different types of variants across the tumor phylogeny compared with solid biopsies (Supplementary Table 11) through the detection of cfDNA-specific subclones (Fig. 8e, Supplementary Fig. 9a–g). These results provide the proof-of-concept for the feasibility of deriving tumor-agnostic comprehensive WGS profiling from a liquid biopsy.

## Discussion

We present a comprehensive technical assessment for cWGTS implementation in clinical care practice in oncology. We demonstrate that using a single integrated workflow, cWGTS captures the full spectrum of cancer-associated genomic alterations that are assessed using a diversity of standard-of-care diagnostic assays. With implementation of best laboratory and computational practices, we execute an end-to-end sample-to-report turnaround time within 9 days, which is aligned to clinical needs for diagnosis and care decisions, and is comparable to infrastructures of scale[18]. Despite 5–10-fold lower sequencing coverage compared with panel-based assays, we demonstrate that in matched biopsies, cWGTS recovered all clinically reported variants by high-depth targeted profiling assays. We establish >80× coverage and tumor purity of at least 20% to attain this sensitivity. However, this sets a stringent quality threshold on

fresh frozen tumor specimens that are not as broadly available as FFPE. However, a major limitation of WGS in FFPE is a high error rate in genome-wide calls[57]. To this end, we provide proof-of-concept feasibility data demonstrating that comprehensive WGS profiling can also be leveraged in patients who have high cfDNA content in circulation at the time of diagnosis or relapse. Our findings pave the way for future studies focused on analytical validation and optimizations of comprehensive tumor-agnostic WGS profiling from cfDNA for diagnostic purposes.

To support cWGTS variant annotation and prioritization, we implemented an analytical workflow that learns from variant annotation databases and integrates signals from germline mutations, somatic DNA, and RNA-seq findings. This allows us to annotate, validate, and prioritize SVs of diagnostic (e.g., *MYB* enhancer hijacking), prognostic (e.g., *ATRX/TERT*), and therapeutic relevance (e.g., *TP53* loss-of-function SVs). Consistent with recent literature for pediatric and rare cancers[7,15,16,58], >50% of patients had additional findings of established biological or clinical significance. The majority of these findings were SVs in cancer genes, fusion genes, and genome-wide mutation signatures that targeted panels are not optimally designed to identify. Importantly, we demonstrate that only a minority of such additional findings would be captured by WES and RNAseq alone or in combination. Larger cohort studies are warranted to determine the incidence and prevalence of clinically relevant biomarkers captured by cWGTS.

The clinical relevance of cWGTS extends beyond that of rare cancers[59]. We show that by cWGTS, we detect the full spectrum of cancer-associated mutations in 99% of patients. The vision of patient-tailored medicine warrants the delivery of clinical decisions that extend beyond a single druggable biomarker and rather consider the composite readouts from a patient's cancer genome that inform on a patient's a priori risk of developing cancer, diagnosis, likelihood of treatment response, risk of progression, and therapeutic vulnerabilities. With increasing implementation of cWGTS on well-annotated clinical specimens[15,16,58], our ability to interpret cWGTS findings will improve, and by extension, the clinical utility of cWGTS will expand. As the economic barriers to cWGTS are mitigated in time, a single comprehensive assessment of the cancer genome is positioned to replace multiple targeted diagnostic tests in prospective clinical sequencing[59].

## Methods

**Study participants**. Patients who were seen within the Department of Pediatrics at Memorial Sloan Kettering Cancer Center with presumed or established solid tumor malignancies (including CNS tumors) were eligible to enroll on an institutional prospective tumor/germline-sequencing protocol (ClinicalTrials.gov number, NCT01775072) with informed consent from the patients or their guardians. This study was approved by the MSKCC Institutional Review Board/Privacy Board. Patients with newly diagnosed as well as relapsed/refractory disease were eligible. Adults with pediatric-type malignancies or rare cancers up to the age of 39 were also eligible to enroll.

**Clinical profiling**. DNA extracted from formalin-fixed paraffin-embedded (FFPE) tumor and blood samples (as a matched normal) was sequenced using MSK-IMPACT, an FDA-approved targeted panel used to sequence patients' tumors at MSKCC. MSK-IMPACT captures protein-coding exons of 468 cancer-associated genes, introns of frequently rearranged genes, and genome-wide copy number (CN) probes[4]. Tumor and normal samples were sequenced at 800× and 600×, respectively. Established pipelines followed by manual review were used to characterize germline and somatic mutations, CN variants, and if targeted, genomic rearrangements as previously described[1]. Germline data for alterations in cancer-predisposition genes were analyzed in 88 genes as previously described[2]. For select tumor indication MSK-Fusion[3], a New York State-approved RNA-capture assay that targets common RNA fusion genes in solid tumors was also performed. Clinically relevant findings were annotated using OncoKb tiers 1–4[4].

**Research-sequencing approaches**

*DNA extraction*. For 114 subjects enrolled in the study, tumor DNA was extracted from fresh frozen (FF) or OCT tissue biopsies and matched normal DNA from

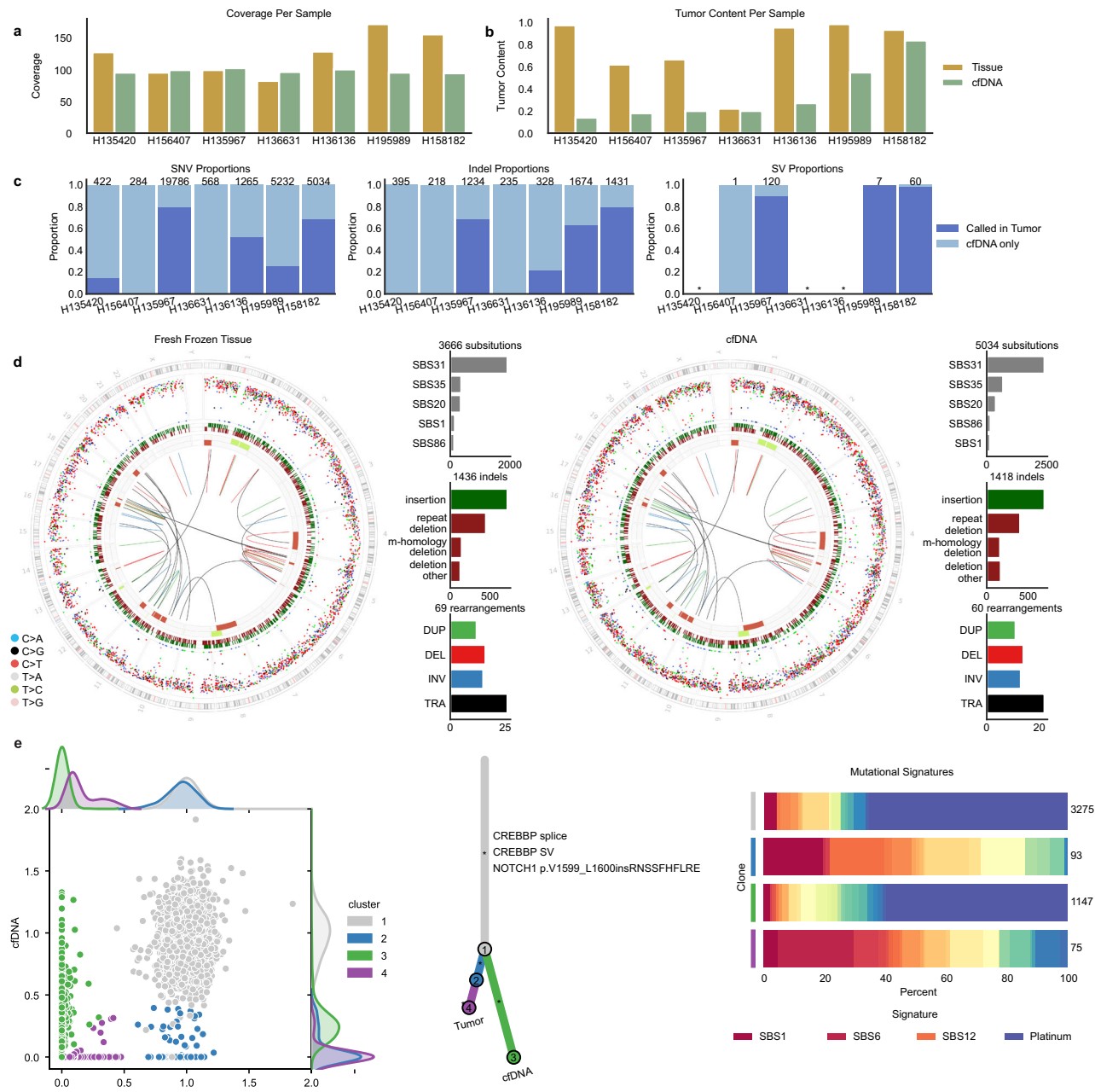

**Fig. 8 Comparison of WGS data from matched fresh frozen tumor tissue and cfDNA. a** Coverage values ordered by estimated tumor context in cfDNA. **b** Estimates of tumor content. **c** Barplots showing the proportion of de novo mutation calls in cfDNA that are present in the matched fresh frozen tumor broken down by variant type. cfDNA samples with no high-confidence SVs denoted with an asterisk. **d** Genome-wide distribution and mutation patterns of matched fresh frozen (left) and cfDNA (right) samples for H158182. Circos plots are shown as described in Fig. 6. **e** Individual-level clonality analysis for H158182. (left) Scatterplot of cancer cell fraction (CCF) values for all substitutions color-coded by the estimated cluster. (middle) Phylogenetic tree representation of clusters annotated with clinically relevant variants. (right) Clone-level mutational signature analysis showing the proportion of mutations attributed to each mutational signature with total numbers of mutations in each cluster shown on the right. Whereas drivers associated with these clones could not be determined, cfDNA-specific SNV calls recapitulated mutation signatures in the FF sample, and were enriched for platinum-associated mutational signatures pointing to the existence of therapy-exposed tumor subclones in circulation. (repeat deletion: repeat-mediated deletion, m-homology: microhomology-mediated deletion, deletion other: all other deletions, TRA: translocation, DUP: duplication, DEL: deletion, INV: inversion). Source data for panels **a**, **b** are provided in Supplementary Data 11. Source data for panel **c** are provided at the data repository. Raw data for panels **d**, **e** can be accessed at the dbGAP study.

buffy coat using the DNeasy Blood & Tissue Kit (Qiagen catalog # 69504) according to the manufacturer's protocol. FFPE tissue was deparaffinized using heat treatment (90 °C for 10′ in 480 µL PBS and 20 µL 10% Tween-20), centrifugation (10,000 × g for 15′), and ice chill. Paraffin and supernatant were removed, and the pellet was washed with 1 mL of 100% EtOH followed by an incubation overnight in 400 µl of 1 M NaSCN for rehydration and impurity removal. Tissues were subsequently digested with 40 µl of Proteinase K (600 mAU/

ml) in 360 µl of Buffer ATL at 55 °C. DNA isolation proceeded with the DNeasy Blood & Tissue Kit (QIAGEN catalog # 69504) according to the manufacturer's protocol modified by replacing AW2 buffer with 80% ethanol. All DNA was eluted in 0.5X Buffer AE.

*Whole-genome sequencing.* After PicoGreen quantification and quality control by Agilent BioAnalyzer, 500 ng of genomic DNA were sheared using a LE220-plus

Focused-ultrasonicator (Covaris catalog # 500569) and sequencing libraries were prepared using the KAPA Hyper Prep Kit (Kapa Biosystems KK8504) with modifications. Briefly, libraries were subjected to a 0.5X size select using aMPure XP beads (Beckman Coulter catalog # A63882) after post-ligation cleanup.

PCR-free libraries were pooled equivolume for sequencing. Samples were run on a NovaSeq 6000 in a 150-bp/150-bp paired-end run, using the NovaSeq 6000 SP, S1, S2, or S4 Reagent Kit (300 cycles) (Illumina). Tumors were covered to an average of 95X (range = 67–181) and normals at 50X (range = 32–159).

*RNA extraction.* Tumor tissue from FF biopsies was homogenized in 1 mL TRIzol Reagent (ThermoFisher catalog # 15596018) followed by phase separation with 200 μL chloroform. RNA was extracted from the aqueous phase using the miR-Neasy Micro Kit (Qiagen catalog # 217084) on the QIAcube Connect (Qiagen) according to the manufacturer's protocol with 350 μL input. Samples were eluted in 15 μL of RNase-free water.

*Whole-transcriptome RNA sequencing.* After RiboGreen quantification and quality control by Agilent BioAnalyzer, 18ng–1μg of total RNA with an RNA integrity number varying from 1 to 9.9 underwent ribosomal depletion and library preparation using the TruSeq Stranded Total RNA LT Kit (Illumina catalog # RS-122-1202) according to instructions provided by the manufacturer with 8 cycles of PCR. Samples were barcoded and run on a HiSeq 2500 in Rapid Mode or HiSeq4000 at PE100 or on a NovaSeq 6000 at PE150, using the HiSeq Rapid SBS Kit v2, HiSeq 3000/4000 SBS Kit, or NovaSeq 6000 SP, S1, S2, or S4 Reagent Kit (300 cycles) (Illumina). Sequencing was performed to achieve a median of 83 million paired reads per sample.

*cfDNA extraction and whole-genome sequencing.* Cell-free DNA (cfDNA) was extracted from plasma using MagMAX cfDNA isolation kit. After PicoGreen quantification, 47–500 ng of cfDNA were used to make sequencing libraries using the KAPA Hyper Prep Kit (Kapa Biosystems KK8504) with 4 cycles of PCR and pooled equimolar. One sample with sufficient input was prepared PCR-free. Samples were run on a NovaSeq 6000 in a PE150 run, using the NovaSeq 6000 SBS v1 Kit and an S4 flow cell (Illumina). The average coverage per sample was 91X.

*Workflow optimization.* In order to achieve stable turnaround times of 9 days, dedicated resources and optimizations were needed, such as to minimize human steps in the process. In the sequencing core, lab technicians along with sequencers were needed to process and quality-control the incoming samples. A high-throughput connection was used to transfer sequencing data to the bioinformatics core with automatic notifications. An ETL cron job was developed to synchronize relevant deidentified metadata regularly from clinical systems. The data and bioinformatics analyses were tracked and automated using the Isabl platform[5]. In order to achieve stable algorithm turnaround times, parallelization was often split by the estimated amount of work (i.e., number of reads; https://github.com/papaemmelab/split_bed_by_reads) rather than genomic length. Processing was performed within a heavily shared internal high-performance computing (HPC) cluster with around 4000 cores. The results were automatically curated and prioritized using both cached databases and live APIs in order to reduce interpretation time.

*Bioinformatic analysis.* Analysis of cWGTS data was executed using Isabl platform[5] and included: 1. data QC; 2. ensemble variant calling for germline and somatically acquired mutations from at least two out of three algorithms run for each variant class; 3. signature extraction (i.e., mutation signatures, microsatellite-instability score, and gene expression); 4. variant classification; and, 5. the generation of a clinical prototype summary report. Briefly, upon completion of each sequencing run, Isabl imports paired tumor-normal FASTQ files, executes alignment, quality-control algorithms, and generates tumor purity and ploidy estimates. For samples with sufficient coverage (>60×) and tumor purity (>20%), ensembl variant calling for each variant class (substitutions, insertions and deletions, and structural variations) is performed. High-confidence somatic mutations were classified with regard to their putative role in cancer pathogenesis and statistical post-processing enables the derivation of microsatellite-instability scores and mutation signatures[6]. RNA-seq data were independently analyzed for acquired fusions and gene expression metrics in a subset (n = 101). For a subset of patients with consent (n = 100) for germline analyses, the normal genome was also independently analyzed.

Clinical relevance of mutations in common cancer genes was annotated using OncoKb, COSMIC, Ensembl Variant Effect Predictor, VAGrENT, gnomAD, and ClinVar databases[725] (refs. [60–63]). Additionally, integration of signals across data modalities (germline, somatic mutations, somatic signatures, CN segments, and gene expression profiles) was executed to further determine the significance of observed events. Population filtering, database comparison, and somatic data integration were performed using methods in accordance with the American College of Medical Genetics and ClinGen Somatic/Germline Data Integration subcommittee[64–66]. Last, the findings were automatically embedded into a single-page summary (.html) report containing high-level clinical data, quality-control metrics, genetic findings, and relevant data-visualization plots (i.e., CIRCOS plots, mutation signatures, and gene expression clustering by tSNE). Putative findings of

clinical relevance identified by WGS and RNA-seq were reviewed by an interdisciplinary team of clinical oncologists, molecular pathologists, and cancer genomics experts. Typically 8–10 cases were reviewed in an hour-long tumor board meeting that was held biweekly. The findings were categorized with regard to their relevance in clinical practice as 1. diagnostic, 2. risk predisposition for germline variants, 3. prognostic, 4. therapy-informing, 5. pathogenic, 6. likely pathogenic, or 7. variant of unknown significance (VUS).

## Pipeline overview

*Whole-genome/transcriptome alignment and quality control.* Whole-genome paired-end reads were aligned to human reference genome (GRCh37d5) using BWA-mem (v0.7.17) as a part of the pcap-core v2.18.2 wrapper (https://github.com/cancerit/PCAP-core)[67]. The wrapper includes marking of duplicates using Picard. Whole-transcriptome sequencing reads were aligned using Spliced Transcripts Alignment to a Reference, STAR (v2.5.4b, https://github.com/alexdobin/STAR) with Ensembl 75 for transcript information[68]. Upon alignment, BAM files for tumor/normal WGS and tumor RNA-seq data for each individual were compared using Conpair[69] in order to detect potential sample swaps and cross-individual contamination. Genome-wide median coverage was calculated using Mosdepth[70] with minimum mapping quality of 20. Tumor purity and ploidy was estimated using Battenberg (https://github.com/cancerit/cgpBattenberg) and somatic substitution calls. Additionally, a quality-control report is generated per sample using MultiQC (v1.9) (https://github.com/ewels/MultiQC) to aggregate alignment and read statistics from FastQC (v0.11.5) (https://github.com/s-andrews/FastQC), Picard (v2.25.6) (https://github.com/broadinstitute/picard), and RNA-SeQC (v1.1.8.1) (https://software.broadinstitute.org/cancer/cga/rna-seqc)[70,20].

*Identification of somatic mutations in whole-genome sequences.* Somatic alterations were detected comparing the tumor against the matched normal for each variant type. All bioinformatic tools were launched using an in-house wrapper. Allele-specific subclonal CN changes were detected using Battenberg (cgpBattenberg v1.4.0) (https://github.com/cancerit/cgpBattenberg)[71]. Single-nucleotide variants (SNVs) were identified using Strelka2 (v2.9.1 with manta v1.3.1), (https://github.com/Illumina/strelka), MuTect2 (gatk:v4.0.1.2),(https://github.com/broadinstitute/gatk), and CaVEMan (cgpCavemanWrapper v1.7.5) (https://github.com/cancerit/cgpCaVEManWrapper)[72–74] Variant post-processing was done using default flags for Strelka2 and MuTect2, while for CaVEMan, cgpCavemanPostprocessing (v1.5.2) was used filtering for sequencing artifacts with > =3 mutant alleles in at least 1% of samples within a panel of 100 unmatched blood normal (https://github.com/cancerit/cgpCaVEManPostProcessing). Small insertions and deletions (indels) were detected using Strelka2, MuTect2, and Pindel (cgpPindel v1.5.4) (https://github.com/cancerit/cgpPindel) and filtered against a panel of 100 unmatched normals[75]. Structural genomic variants (SVs) were identified using SvABA (~v1.0.0 commit 47c7a88) (https://github.com/walaj/svaba), GRIDSS (v2.2.2) (https://github.com/PapenfussLab/gridss), and BRASS (v4.0.5 with GRASS v1.1.6) (https://github.com/cancerit/BRASS) using a panel of 100 in-house unmatched normals[76,77].

Finally, microsatellite-instability status was assessed using MSISensor (v0.5) (https://github.com/ding-lab/msisensor)[78].

*Variant consolidation and annotation.* VCF files for SNVs and indels were merged with an in-house wrapper using chromosome, position, reference allele, and alternative allele. The merged VCFs were annotated with VAGrENT (v3.3.0, https://github.com/cancerit/VAGrENT) and VEP (v92, https://github.com/Ensembl/ensembl-vep)[63,79] VCF files for SVs were merged using MergeSVvcfs (v1.0.2, https://github.com/papaemmelab/mergeSVvcf). High-confidence mutations were designated as those that were passed by at least 2 callers.

*Designation of putative oncogenic mutations.* We define a variant as oncogenic if it represents an established "driver" mutation on the basis of prior literature and recurrence in cancer genome. For SNVs, indels, and fusion genes, these annotations are derived from OncoKb. For SVs, we annotate events that target known oncogenes and tumor suppressor genes and use prior literature as reference (e.g., for TERT, ATRX, and TP53 SVs).

*Identification of germline mutations.* Germline single-nucleotide polymorphisms (SNPs) and indels were detected using Stelka2 and Freebayes (v1.2.0, https://github.com/ekg/freebayes) with an in-house wrapper. VCF files were merged and annotated using the same procedure used for the somatic variants[80]. Germline variants called by both callers were considered high-confidence. Germline variants were prioritized for review by filtering for recurrence in the current cohort, frequency in any population of 1000 genomes/Gnomad and ClinVar.

*Characterization of gene fusions.* Gene fusions were identified using three different callers: FusionCatcher (v1.0.0, https://github.com/ndaniel/fusioncatcher), STAR-Fusion (v1.3.1, https://github.com/STAR-Fusion/STAR-Fusion), and FuSeq (v1.1.1, https://github.com/nghiavtr/FuSeq)[81–83]. Calls were merged by gene pair and annotated using FusionCatcher's databases. Fusions were considered confident if

called by at least 2 callers. Events were visualized with the plotting functionality by Arriba (https://github.com/suhrig/arriba)[84].

*Gene expression analysis.* Gene expression profiles were ascertained in transcripts per million (TPM) using SALMON (v0.10.0, https://github.com/COMBINE-lab/salmon)[85]. tSNE was performed on RNA-seq data from 101 tumors from the cohort and an in-house reference consisting of 155 pediatric tumors using python scikit-learn (v0.21.1, https://scikit-learn.org/) and visualized interactively using python bokeh (v1.2.0, https://docs.bokeh.org/)[86].

*Identification of gene expression biomarkers.* Expression biomarkers were assessed using the methodology outlined by Horak et al.[16]. Only actionable genes outlined in the publication supplementary as assessed by severe overexpression, overexpression, severe underexpression, or underexpression, were evaluated. An internal reference cohort of 274 tumor RNA samples was used as a baseline. A gene was considered over-/underexpressed if expression was in the top or bottom ten percent of the reference cohort, while severe over-/underexpression was categorized as expression in the top or bottom five percent.

*Calculation of tumor mutational burden.* Tumor mutational burden (TMB) was calculated using high-confidence, somatic substitutions and indels that fall within coding regions. The totals for these variant classes were combined and then converted to coding TMB using a divisor of 30 to approximate the length of the human exome in Mb. Values greater than 2 coding mutations per Mb were considered pediatric high and values greater than 10 coding mutations per Mb were considered hypermutators, thresholds set by the study in Grobner et al.[16].

*Identification of mutation signatures of point mutations.* Mutational signature analysis was performed with the MutationalPatterns package (v1.6.1, https://bioconductor.org/packages/release/bioc/html/MutationalPatterns.html) and using the COSMIC Mutational Signatures (v3.1) with the addition of Temozolomide signature from Kucab et al.[87,88].

*Assessment of ITH between matched MSK-IMPACT and WGS samples.* ITH between the matched FFPE and FF samples that underwent MSK-IMPACT and WGS sequencing was assessed by comparing CN changes and substitutions/indels falling within 468 genes included in MSK-IMPACT. For substitutions/indels, the clonal representation in the two assays was compared by performing a proportion test comparing the VAFs reported and adjusting for assay-specific local depth and purity. Mutations with a p-value <0.05 have a statistically significant difference in clonal presentation suggesting ITH. CN profiles from MSK-IMPACT generated using FACETS[89] were compared with the Battenberg output from WGS. In patients where DNA was available, resequencing of discordant mutations was performed using the MSK-IMPACT panel on the same DNA that underwent WGS sequencing at a median depth of 438X[90]. Tumor purity of both assays was taken into account to mitigate the effects of technical issues on mutation calling.

*Inference of clonal structure.* Clonal structure was analyzed using high-confidence SNVs called in each biopsy or the union of SNVs whenever multiple biopsies were available for a patient. DPClust (v0.2.2, https://github.com/Wedge-Oxford/dpclust) was used for calculation of cancer cell fraction corrected for purity and local CN, as well as clustering and assignment of mutations across samples with the exception of the Gibbs Sampling Dirichlet Process step that was optimized internally[71]. Clonal ordering was deduced using clonevol (v0.99.11, https://github.com/hdng/clonevol)[91]. Mutational signatures were computed in each cluster independently. Figures were generated with matplotlib (v3.1.0, https://matplotlib.org/).

*Estimates of telomere length.* The ratio of telomere length in tumor vs normal was estimated using Telseq (v0.0.2, https://github.com/zd1/telseq)[92].

*Derivation of subsampled BAM files and sensitivity assessment.* A total of 298 subsampled BAMs were generated using samtools (v1.11, https://github.com/samtools/samtools) view command with the subsampling option[93]. Median coverages were calculated for the original BAMs using Mosdepth (v0.2.5, https://github.com/brentp/mosdepth) with mapping quality >20 and then used to calculate fractions to downsample to approximately 100×, 80×, 60×, and 30×–40× where original coverage was allowed[70]. Mosdepth was used to verify that the median coverage of the subsampled BAM fell within +/−5× of the desired coverage. De novo variant calling and annotation was then performed independently on the subsampled BAM files using the same procedure as cWGTS as described above.

*cfDNA tumor content and variant comparison.* Tumor content in cfDNA specimens was estimated using Battenberg and manual inspection with the help of SNV VAF density plots. De novo variant calling was performed independently using the methods described for identification of somatic mutations in WGS. Further analysis was done to compare specific clinical variants identified by MSK-IMPACT using hileup (v1.0.0, https://github.com/brentp/hileup). Clonal structure across tissue and cfDNA samples was inferred using the same methods as described before followed by analysis of clone-specific mutational signatures. All mutations across the clonal structures were then piled up across corresponding FF WGS data to look for evidence of mutations in both specimens.

*Variant curation and characterization of incremental findings from cWGTS.* Variant curation of targeted NGS assays was performed as previously described[4,28]. To assess variants identified by cWGTS, a multidisciplinary team of disease experts, clinical geneticists, molecular pathologists, and genomics experts assembled regularly to classify molecular alterations (somatic and germline), mutation signatures, and gene expression data. Incremental findings of cWGTS were defined as established oncogenic alterations or signatures not identified by matched MSK-IMPACT somatic or germline NGS (DNA) or ArcherDx targeted NGS (RNA). Incremental findings were further classified as clinically relevant if they met one of the following criteria: (1) diagnostic finding—defined as an alteration that provided justification or an alteration in cancer diagnosis or cancer-subtype diagnosis, (2) established prognostic finding—defined as an alteration with established prognostic relevance with robust support from scientific literature, (3) likely pathogenic or known pathogenic germline- predisposition event, (4) treatment-informing finding—defined as an alteration that provides direct justification of a therapeutic modality, or (5) a driver oncogenic fusion.

**Reporting summary**. Further information on research design is available in the Nature Research Reporting Summary linked to this article.

## Data availability

The raw data for WGS and RNA-seq data generated in this study have been deposited in the dbGAP database under accession code phs002620.v1.p. These data are available under restricted access due to individual privacy concerns. Permanent employees of an institution at a level equivalent to a tenure-track professor or senior scientist with laboratory administration and oversight responsibilities may request access through dbGAP. The requests, which are managed by NCI's Data Access Committee, take less than 2 days for approval and access is permitted for 12 months. The processed MSK-IMPACT data are available in a study-specific dataset at cbioPortal [https://www.cbioportal.org/study/summary?id=mixed_kunga_msk_2022]. Summary and processed data for the figures are available in the source data file as well as the data repository at https://github.com/papaemmelab/Shukla_Levine_Gundem. Annotation databases included public resources such as Cancer Gene Census, OncoKb, ClinVar, 1000genomes, gnomAD, and Ensembl Variant Effect Predictor (VEP) databases. The remaining data are available within the article and Supplementary Information file.

## Code availability

Scripts for generating the figures are provided at https://github.com/papaemmelab/Shukla_Levine_Gundem.

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

## Acknowledgements

The authors would like to acknowledge Drs T. Heaton, M. LaQuaglia M, and J. Gerstle for executing surgical resections, Drs N. Schultz, D. Chakravarty, and M. Nissan for assistance with OncoKb annotations, Drs. D. Solit and A. Kentsis for support and interesting discussions, and Drs. N.K. Cheung, B. Kushner, E. Basu, P. Meyers, L. Wexler, P. Kothari, I. Dunkel, S. Gilheeney, Y. Khakoo, K. Kramer, S. Prockop, and T. Trippett for participation and clinical contributions to the study. E.P. is a Josie Robertson Investigator and is supported by the European Hematology Association, American Society of Hematology, Gabrielle's Angels Foundation, V Foundation, and The Geoffrey Beene Foundation, and a Damon-Runyon Rachleff Innovator Award recipient. Funding for this study was supported by the Olayan Fund for Precision Pediatric Cancer Medicine.

## Author contributions

E.P., A.L.K., and N.S. designed the study. M.F.L, G.G, D.D. J.G.A, J.S.M.M, Y.Z, and J.E.A.O developed algorithmic infrastructure and M.F.L, G.G, D.D, D.G, J.G.A, and JEAO performed bioinformatic analysis. N.S., N.B, E.S, L.S, I.S.R, and T.O oversaw coordination of patient consent, sample processing, and clinical data acquisition. U.B, M.H.A.R, and S.S.F oversaw biospecimen banking and pathology review. N.S, M.F.L, G.G, B.S, D.G, E.F, J.B, T.O, F.I.C, M.O, M.K, S.R, S.M, E.S, M.K, M.L, F.D.C, J.G.B, A.Z,M.F.W, A.L.K, and E.P oversaw variant annotation and data interpretation. C.C and A.V executed laboratory processing of biospecimens and sequencing. M.F.L, G.G, D.D, and E.P prepared figures and tables. N.S, G.G, A.L.K, and E.P. wrote the paper with input from M.F.L, D.D, and M.F.W. All authors reviewed and approved the paper for submission.

## Competing interests

E.P., A.L.K and J.S.M.M are founders, equity holders and hold fiduciary roles in Isabl Inc. E.P., A.L.K , J.S.M.M, M.F.L. G.G, J.Z, J.E.A.O, J.G.A, N.S are inventors on intellectual property related to a software platform for genomic data analytics, a non-provisional patent application has been filed 17/629292 titled "SYSTEMS AND METHODS FOR CANCER WHOLE GENOME AND TRANSCRIPTOME SEQUENCING (CWGTS)". G.G. is a consultant in Isabl Inc.D.G. is a consultant and a shareholder of Repare Therapeutics. D.G. is a consultant and stock-option holder in MNM Diagnostics.
