## [Peer review file · Nature Communications]

REVIEWER COMMENTS

Reviewer #1 (Remarks to the Author):

The authors have satisfactorily addressed all concerns in this reviewer's opinion. This is a well done genomic study that warrants publication.

Reviewer #4 (Remarks to the Author): Expert in cancer genomics

Review Shukla et al.

Lines 77-80. This is largely incorrect. FFPE material and cfDNA suffer from the same low purity issue (in combination with WGS). The solutions implemented for these input materials (i.e. panel-based deep sequencing) are equally applicable to FF material. So in my opinion this sentence should convey the message that more sequencing depth is required for lower purity samples, which is currently not economical, but with decreasing sequencing prices may become feasible. Alternatively, (laser-capture) tumor enrichment and/or panel-based approaches would address the challenge. I don't think FFPE or cfDNA do make sense to bring up here as a solution for low purity issues.

Line 92/93. This is not correct. See [https://www.jmdjournal.org/article/S1525-1578\(21\)00120-3/](https://www.jmdjournal.org/article/S1525-1578(21)00120-3/) reporting 10 days. In general, proper reference to and comparison with the results in this paper are largely lacking in the current manuscript.

It is also not clear if for all 114 patients WTS data was successfully generated (and with what depth and QC metrics). Statistics on the success rates for this should be added to the sample processing section.

Figure 1B is misleading. Best reported turnaround time is not very informative from a clinical perspective. This graph should be converted to median reported turnaround time (preferably with ranges). Own data about the four consecutive batches (line 89) should all be included in this figure to highlight the speed and impact of learning curve and optimization.

More than 20% of variants are discordant between MSK-IMPACT and WGS and are explained by ITH. This number is quite high. Is this because primary tumor samples were analyzed, which are known to be more heterogeneous than metastases? Were any metastases included and was the fraction of discordant observations similar? The main body of the text does not include information on the source of the material analyzed and stage of the disease. It would be useful to add such information to be able to better interpret the ITH numbers.

The statement in line 135 (cWGTS had 100% concordance) is misleading. This is not proven because only 44 out of 47 discordant variants were followed up. You can only state: at least 99% or something. This statement is misleading in second way as the comparison should always be done in both directions to conclude something about concordance. How many variants were detected by WGS in the target regions of MSK-IMPACT that were not seen/called by MSK-IMPACT (potential false positives by WGS or false negatives by MSK-IMPACT). This analysis should be added to the manuscript to make concordance claims solid.

And as a minor comment, cWGTS should read WGS in this sentence as no RNA data was used here.

The added value of WTS is poorly described. Did WTS yield more relevant biomarkers than the 18 fusions? As these were all also detected by WGS, shouldn't you conclude that the added value of WTS is rather limited from a diagnostic perspective? When the WGS findings are solid and validated, there is formally no need for an independent orthogonal validation.

The section on integration of RNA-seq and WGS is largely descriptive and is insufficiently quantitative to judge the added value of RNA-seq for diagnostic testing. I would like to note that I do not want to deny the value of WTS in general, but it's utility in the clinic is currently largely anecdotal and not coupled to formal actionability as far as I am aware.

I think this section would benefit from a clear split (table?) in numbers of cases where WTS alone provided actionable information, where WTS supported the WGS results and where a conclusive outcome was only possible based on WGS and WTS. This quantification is highly relevant as choosing for WGS or WGTS also needs a balance between added value and added costs, which is not yet easily possible with the data currently provided.

The additional structural variants targeting tumor suppressor genes could benefit from some more details on the nature of the SVs. Are these all (small) deletions, or just gene-disrupting break-junctions, or part of complex events, etc. This helps understanding why these are additional WGS findings

WGD rates are much lower than reported in <https://www.nature.com/articles/s41586-019-1689-y>. Is this due to differences in tumor type distribution or does this relate to primary versus metastatic lesions?

TMB estimates were overall 8.3 fold higher than previous reports. This needs some explanation. I guess previous studies were panel of WES-based. I would be good to also report on the coding TMB as it is known that mutation rates differ between coding and non-coding sequence. Could this at least partially explain the difference?

The cfDNA analyses on only seven patients is insufficiently powered to be informative and does not add to the main message of the manuscript or the data already present in scientific literature (e.g. <https://www.nature.com/articles/s41467-021-23445-w>, which provides much more solid numbers). I believe this part would better be removed from the manuscript.

The reported cfDNA percentage is 10 to 83%. This appears extremely high compared to numbers that are typically reported in literature (e.g. median of only 0.18% in this study: <https://www.nature.com/articles/nm.1789>). This needs additional details on how these percentages were achieved and determined and a note that these purities are very atypical (and therefore the inclusion of a statement on cfDNA analysis feasibility in the abstract is not justified). What type of tumors had these patients?

Line 320: the lower limit of ctDNA should be mentioned explicitly.

A statement on data availability is lacking.

Related to the reviewer comments

R1, Comment 1: the procedures detailed here for clinical decision making and tumor board support should be included in the manuscript.

R1, Comment 3: the statement on 'the ability to derive high quality WGS from cfDNA has never been demonstrated...' is not true: see <https://pubmed.ncbi.nlm.nih.gov/30617129/> or <https://www.nature.com/articles/s41467-021-23445-w>

R3, Comment 2: this is a main concern that I share and which is not properly addressed in the current version of the manuscript.

R3, comment 6: the reviewer asks for TAT overview of all samples over time. This is still not provided in the revised figure.

R3, comment 5 and 9: I share these concerns regarding the demonstrated added value of WTS. This is not sufficiently addressed in the current version of the manuscript. See also my comments above.

R3, comment 15: the involvement of structural variants (e.g. LOH) resulting in almost 100% biallelic inactivation of TP53 rates

Reviewer #5 (Remarks to the Author): Expert in paediatric cancer genomics

The manuscript prepared by Shukla and colleagues describes the development and implementation of paired genome sequencing and transcriptome sequencing in prospective institutional cohort of adolescent young adult and adult patients <40y with newly diagnosed, treatment refractory or relapsed cancer. A major focus of the manuscript is the feasibility of translation from a research environment into a clinical setting.

The concepts presented herein are not novel in the sense that cancer-based exome, genome and transcriptome sequencing have been previously well-described in the literature, including validation of these methods in a clinical setting. The authors give appropriate reference to those prior studies. Areas highlighted in the manuscript include not only the expected SNV, CNV and SV/fusion detection performance, but also extend into the extraction of mutational signatures, microsatellite instability, chromothripsis, and telomere length, among others. Furthermore, the manuscript includes discussion of quality metrics for assay performance and the use of the applied bioinformatic tools in this space. Commentary and data surrounding optimal depth for genome sequencing is of great value to the reader and well described.

Overall, given the clinical focus, the manuscript would benefit from providing greater detail in the methodologic approach to genomic results interpretation and reporting. A major emphasis of the manuscript is the ability to turn around genome and transcriptomic results in 9 days. However, the very limited sample set (n=16) and time period in which this was employed speaks more to a theoretical timeline rather than a routine and established workflow. Cell-free DNA is conceptualized toward the end of the manuscript, however the very limited study (n=7), and the variability in sample preparation makes it difficult to draw clear conclusions from these data.

1. Limited detail is provided on how variants were classified into the stated categories below and what attributes related to a variant were considered. Please expand on the decision-making process and

definition of the categories below. Also, can the authors comment on why the AMP/ASCO/CAP guidelines for somatic variant interpretation not employed?

1. Diagnostic 2. Risk predisposition for germline variants, 3. Prognostic, 4. Therapy informing, 5. Pathogenic, 6. Likely Pathogenic or 7. Variant of unknown significance (VUS).

2. While cWGTS analyses extended the detectable variant and signature spectrum, the description of the extension of findings is challenging to follow and difficult to discern from Figure 4a and Supp Table 2. This is particularly true when referencing that “less than half of the findings in cWGTS could be captured by WES and RNA-seq” (Line 176, page 8). Recommend to add to the text that the majority represented SV, and to clearly indicate in a column in Supp Table 2 which findings were detected by panel sequencing and which would have been captured by exome and RNA sequencing. Also, it is of interest for the reader to know if other well-described and clinically relevant genomic alterations were present in these exact samples which would have been captured by exome and RNA seq or if this new finding represented the only disease-associated finding of relevance. This speaks to understanding how broadly use of cWGTS extends the analysis for clinically relevant findings, vs adds to an existing broad repertoire of detected variation in a given sample using existing well-described techniques.

3. Clear emphasis in the manuscript is focused on a 9 day TAT for cWGTS. Given the limited sample set (n=16) and restriction to only 4 batches, the time of the established workflow should be de-emphasized. Recommend to report the TAT from across the entirety of the study which seems to be more in fitting with allowing a direct comparison to other published studies. Additionally, a critical missing component is the time spent on review/ interpretation of genomic data with the interdisciplinary team who is establishing relevance in clinical practice and the frequency of such meetings.

4. Could the authors comment on employed quality metrics other than depth and tumor content and how those were determined? Is a QC report generated for each patient, and if so which data are encompassed in that review? Additionally, can the authors provide data on the reduction in spurious variant calls through use of CaVEMan as related to the ability to filter out sequencing artifacts. Finally, can the authors comment on how a filtration strategy was applied in the panel of 100 unmatched normals (are these matched by tissue type- blood vs FFPE vs FF), and how many artifacts were eliminated via that approach? Understanding the initial annotated variant list followed by filtration/prioritization, and at the end, the range of variants considered in the context of clinical review would aid the reader.

5. Define the term “oncogenic” which is used throughout (e.g. page 6, line 132; Figure 4a)

6. In reference to a germline alteration, refrain from using the term mutation, as variant has become the standard term in the field. Please update throughout. Additionally, in the methods section under the

heading “Identification of Germline Mutations”, the term SNP is used. Please update to SNV as germline disease-associated variants are not polymorphisms.

7. A drawback of this study is limitation to fresh, frozen tissue with a significant level of attrition due to failure to meet tumor purity requirements. Can the authors comment on the type of biopsies (core needle vs resection, etc) and the typically provided size/weight estimates of the available tissue to provide the reader greater clarity on the nature of the source under study.

8. Were orthogonal methods applied to validate the signatures data sets generated from cWGTs? For instance, how was accuracy in telomere length assessed? This is of interest given the very modest association with length relative to mutational status (Supp. Figure 7).

9. Please provide greater detail on the calculation employed for TMB and the definitions of hypermutator and pediatric high referenced in the supplemental.

10. Among the variants unique to MSK-IMPACT, was manual review of the panel and genome data performed to discern if there was an underlying contributor such as strand bias, alignment issues, base quality, repetitive sequence, or other cause as opposed to intratumor heterogeneity (Line 130, page 6)?

11. Was erroneous alignment ruled out as a contribution to the difference between the HLA-A call between genome and targeted MSK-IMPACT sequencing (page 6, line 132)?

12. Supplemental Figure 6 seems uninformative in that similarly described cancer histologies fail to cluster discretely in the supplied TSNE. Can the authors comment on why this is the case? Perhaps the categories are too broad with variability in the underlying driver alteration? It is also not clear what constitutes the in-house reference cohort.

13. References 1 and 2 are identical in the methods, as are references 5 and 8. Please check references throughout for accuracy.

14. S.Figure1 a should have a legend to describe the disease type displayed in the chart.

15. Supplementary Table 1, convert age from days to years.

As requested by the editor, this portion of the review serves as an assessment to the degree to which the authors have responded to the comments from Reviewer 3. Prior to each point, this reviewer's conclusion surrounding the response is listed, followed by Reviewer 3's original comment and the author response.

Comment 1 response:

I am in agreement with reviewer 3, that the authors still substantially focus on somatic genome data, rather than explicitly emphasizing the value of RNASeq and germline components. As related to germline, the authors highlight extension of detection of genome sequencing in picking up 4 germline variants not detected by the MSK-IMPACT panel. Exome would certainly be anticipated to pick up these variants, therefore the lack of coverage due to the design of the MSK-IMPACT panel which is declared as standard of care by the authors is a moot point. The limited data presented herein do not extend the utility of genome for germline variation.

Additionally, as related to the transcriptome, it is unclear how Supp. Figure 6 and Supp. Table 9 extend valuable information to the reader. Supplemental Figure 6 seems uninformative in that similarly described cancer histologies fail to cluster discretely in the supplied TSNE. Perhaps the categories are too broad with variability in the underlying driver alteration. It is also not clear what constitutes the in-house reference cohort. A UMAP may provide more meaningful information, with regard to validity in clustering of similar tumor types. Supp. Table 9 contains information of relevance, but this should all be contained in a master table which displays the tumor/patient identifier, clinically relevant germline and somatic SNVs/indels, oncogenic germline and somatic SNVs/Indels, fusions, and expression outliers for ease of a reader in being able to understand if altered expression is correlated with genomic findings, vs consistent with a given tissue/tumor type.

Comment 1:

The overall impression of the manuscript is that it largely achieves the stated goal of supporting the clinical utility and feasibility of WGS for identifying somatic alterations and mutational signatures, but that it gives little to no attention to these areas for germline susceptibility or for the impact of the transcriptome.

We thank the reviewer for raising this concern. Indeed, evaluation of the germline and transcriptome were integral components of our study, although we did not dedicate specific sections of the manuscript to these categories but rather presented germline and RNA-seq results within the integrated subheadings that formed the narrative of the paper.

Specifically, we incorporate results of the germline analysis in three sections: First, we describe concordance of germline findings by cWGTS to established clinical assays in results section

“Concordance analysis of cWGTS to targeted DNA and RNA panel tests”. We then describe novel germline findings in the results section “Additional findings of biological and clinical relevance detected by cWGTS” in the rare variants section. Last, we dedicate a results section on germline variant annotation and reporting using information from somatic mutation signatures.

Likewise, for the RNA analyses we provide concordance data to MSK-Fusion in results section “Concordance analysis of cWGTS to targeted DNA and RNA panel tests”, we then describe novel fusion findings in the results section “Additional findings of biological and clinical relevance detected by cWGTS” in the fusion results section, and showcased how RNA-seq data (fusion genes and gene expression) were used to further annotate and prioritize putative oncogenic variants identified by cWGTS.

To address the reviewers concern on global RNA-seq analysis we have now introduced a results section titled “Integration of RNA-seq and WGS for variant annotation” highlighting the following three aspects of RNA-seq data integration:

- Validation, annotation, and prioritization of novel fusion genes, and non-coding structural variants.
- Global gene expression clustering to inform on disease classification (S.Figure 6).
- Evaluation of gene expression biomarkers of putative therapeutic relevance (S.Table 9).

Comment 2 response:

The authors partially address comments as related to reviewer 3’s concerns. I am in agreement that focus could improve this manuscript further. For example, at times the authors compare results of genome sequencing against that of a targeted panel test, which they consider to be standard of care. This is one institution’s standard of care. It is of greater relevance to compare against an exome which is increasingly being utilized in a clinical setting. While comparison against an exome is performed for some portions, and should be the benchmark for which increased yield as related to the genome is standardized.

Additionally, there is limited value in providing information in this manuscript as related to cell-free genome studies. It would seem better to have a separate short report on those findings given the very limited sample size and the high degree of discordance. This will clearly become a valuable and worthwhile cohort to report on with regard to cell-free studies, however presently the small n limits interpretation and makes this portion lacking clarity for the reader. Also, Supp Table 10 is missing a Diagnosis field in column E. Additionally, check on the ability to retain full dates as related to procedures (re:HIPAA) in columns C and D. If retaining this dataset, consider updating to a span of days.

Comment 2:

In fact, there seems to be some "drift" from the stated goals of the study that are emphasized in the abstract when compared to what is presented in the text, including a brief foray into comparing primary to recurrent cancers, as well as a very small number of samples evaluated through WGS of circulating free DNA when compared to the WGS analysis results of the tumor DNA. As such, it seems the manuscript needs to be more focused in order to effectively support the clinical utility and feasibility goals that are stated.

We would like to clarify that while our cohort consisted of patients at different points of their treatment journey, contrary to the reviewer's interpretation we only analyzed a single sample per patient and no comparison between primary and recurrent disease was presented. To avoid confusion, we have now included the following statement in the cohort description:

The final cohort included a single sample each from 114 pediatric adolescent and young adult patients (median age = 12.6 years, range: 4.5 months to 43.8 years) with solid tumors (S.Table 1 and S.Figure 1a-d).

The manuscript indeed focuses on the technical considerations for the clinical implementation of cWGTS, providing:

- A head-to-head comparison of cWGTS analytical sensitivity against state-of-the-art panel testing for germline, somatic mutations and fusion genes.
- A benchmarking study to inform technical considerations for cWGTS implementation that consider tumor purity, variant clonality, and sequencing depth.
- Comparison of clinical findings between panel-based assays, whole exome sequencing, RNA sequencing and cWGTS assays.
- A detailed evaluation of the need to integrate germline WGS, somatic WGS and RNA-seq findings for variant annotation, prioritization and reporting of cWGTS biomarkers evidenced with specific clinical case examples.

With regards to the small numbers for the cfDNA cohort, we acknowledge that these are early exploratory data meant to consider alternative sources for expanded genomic analysis. The cfDNA section demonstrates for the first time the possibility to extend WGS applications to cfDNA, and that

when sufficient cfDNA is present, these can deliver full coverage comprehensive WGS profiling across variant classes. The ability to derive high quality WGS data from cfDNA has not been demonstrated before and represents a major advancement in

our field and a novel aspect of the manuscript. We believe that understanding the sample types that can be used for comprehensive genomics is critical for translation of WGS to the clinic and an important addition to the manuscript.

To address the reviewer's concern, we have now highlighted the need to improve applications of WGS in the results section "Sample processing", and unified these thematic observations in the discussion as follows:

"However, this sets a stringent quality threshold on fresh frozen tumor specimens which are not as broadly available as FFPE. However, a major limitation of WGS in FFPE is a high error rate in genome wide calls⁴⁹. To this end, we provide feasibility data demonstrating that WGS profiling can be leveraged in patients with limited surgical biopsies, but who have high cfDNA content in circulation. Future optimizations on fresh frozen biopsy preparation and cfDNA variant calling methodologies will enable expansion of WGS approaches in clinical oncology."

Comment 3 response:

This has been satisfactorily addressed.

Comment 3:

Ironically, the manuscript title leaves out the transcriptome, only listing that 'whole genome profiling' will be examined in these contexts.

We appreciate this comment and agree with the reviewer. We have now changed the manuscript title to: "Feasibility and clinical utility of whole genome and transcriptome profiling in pediatric and young adult cancers. "

Comment 4 response:

The authors have partially addressed reviewer 3's concerns on figures. In addition to the below:

- Define “small mut” in Figure 4b as that term is not used elsewhere in the paper.
- In Figure 5a define genomic reference used as related to breakpoints provided.
- Define “TRA” in Figure 5e as that term is not used elsewhere in the paper.
- In Figure 5c legend, is this meant to say chromosomes 6, 9 and 18, rather than 6, 8, 18? I am thinking that is the case based on the chromosomes highlighted relative to your breakpoints.
- Figure 6 legend and throughout the paper/tables, used proper HGVS nomenclature to describe splice variants.
- Figure 6d, recommend to define what comprises categorical definition of the 89064 indels and 568 rearrangements. For example, what is “deletion other” or “m-homology”, or “t. duplication”?
- S.Figure1 a should have a legend to describe the disease type displayed in the chart.
- S. Figure 7 e, for chromothripsis, define SV vs disruptive vs fusion

Comment 4:

Finally, the figures are entirely too multi-part and in some cases, are poorly described by the figure legends including several that do not even have a title listed for the figure.

We thank the reviewer for this concern and agree with the oversight on our part to optimally describe our figures. To address the reviewers concern we have now performed the following actions:

1. Added a title to every figure that emphasizes the thematic link between each of the panels.
2. Provided more detailed descriptions for each results panel.
3. Removed panels (1b, 2f, 6a,b,e) from the main figures and split figure 2 into two separate figures to minimize load.
4. Re distributed the content of the supplementary figures and extended these to now include 10 S.Figures as compared to 7 in the original submission.

Comment 5 response:

This is partially addressed. Recommend to extend on the history here. Because pediatric cancers have some fundamental differences to those in the adult setting, more broadly applied studies of exome and/or transcriptome sequencing have been initiated for some time (Parsons, Baylor, BASIC3; PipSeq Columbia; St. Jude, PMID: 30262806).

Comment 5:

The Introduction is brief and basically ignores the specific aspects of the pediatric/AYA cancer genome that make DNA plus RNA profiling so critical: common fusion drivers, low mutational burden, methylation-driven gene expression profiles. Fine-tuning this a bit to the challenges might support the notion of broader profiling of DNA and RNA a bit better as a precursor to presenting the results.

We agree with this point. We have now incorporated this specific point in the introduction through the inclusion of the following section:

“The introduction of next-generation sequencing (NGS)-based targeted gene panel assays has aided disease diagnosis, guided care, and improved patient outcomes through refinement of treatment options^{1–5}. However, targeted panels are optimized to assess clinical biomarkers in common cancers^{1,24}. In contrast, for patients with pediatric or rare cancers that have low mutation burden and are primarily driven by structural variants and fusion genes, panel tests fail to identify a clinical biomarker in most cases ^{1–4}. This underscores an unmet need for better diagnostic workflows to guide clinical management. “

Comment 6 response:

The response by the authors does not fully address the highly valid comments made by Reviewer 3. The presented TAT data are highly stylized through the use of only 4 cases processed per week across a 4 week period (n=16). I am in agreement that presenting TAT across the initiation of testing in this cohort would be very meaningful to the reader. If a retrospective analysis is not possible, are there additional recent data which can be added which support sustained TAT of this nature? Additionally, if the goal is for a clinically relevant TAT, it would be meaningful to include the time period between the interdisciplinary meetings and final, fully interpreted result.

Comment 6:

In the results section, the timeline to results aspect is presented briefly for the 114 cases, yielding a best case scenario of 9 days to results, as assayed (apparently) during a 4 week period when there were 4 cases per week being profiled. Ostensibly, this was done once all the challenges of implementing the different 'moving parts' of the workflow were addressed, and represents a best-case scenario, which is fine. However, it would be more informative to present TAT during the interval over which all 114 patients were studied, discussing the range of TAT that was experienced, how that improved over the interval of the study being reported and what innovations or developments most significantly impacted the TAT. In particular, since the ISABL pipeline/process has already been reported, it would help to understand how profound its impact was on the overall TAT. I would basically suggest replacing the non-

informative Figure 1a with a more realistic representation of the changes to the pipeline over time and their impact on TAT.

A major goal of our work was to devise a workflow and analytical pipeline that could achieve a clinically relevant TAT. This was achieved through optimization of every step and was not attributable to a single implemented solution. Moreover, development and optimization of the workflow was iterative and not a discreet stepwise implementation for which we could measure performance changes before and after a specific point in time. Once we locked down our workflow, we executed the time trial over 4 weeks utilizing our now standard throughput of 4 patient cases per week, thus establishing the TAT for our optimized and now standard workflow/throughput.

To address the reviewers concern we have now revised Figure 1a to include a detailed overview of the key dependencies and requirements for the 9 day turn around period and how this compares with the most recent published literature. We have also expanded the relevant

results section to highlight the key resources that were required to execute upon this timeline as follows:

‘To prototype a clinical cWGTS workflow, we developed an end-to-end process (Figure 1a) that included dedicated: 1. Project management team; 2. Lab operators for sample processing; 3.

Sequencing machines for cWGTS; 4. Data import channel; 5. Biosciences platform for automated deployment of analysis pipelines and API integration with institutional and public databases¹⁶; 6. Reserved computing nodes in a high performance computing environment; and, 7. Systematic pipeline for prioritization and reporting of genomic findings (Figure 1a).’

Last, we provide more specific details in the Supplemental Methods section Workflow Optimization.

We would also like to clarify that the published Isabl platform (Medina et al, BMC Bioinformatics) refers to the platform designs that enable automated execution of complex analytical workflows rather than the clinical workflow (sample processing, sequencing, analysis, variant annotation and reporting) described in this manuscript.

Comment 7 response:

This has been satisfactorily addressed.

Comment 7:

Figure 1b illustrates that many of the 114 tumors appear to be recurrent tumors (based on the "treatment signature" designation. Based on later description of treatment signatures, it appears that many of the 114 samples are primary:recurrent pairs of tumors. I think this is an important fact that needs to be stated in the first section of the results so it is clear to readers.

As in the response to Comment 2 above, we would like to clarify that the study did not include primary:recurrent pairs but rather included patients with either primary or recurrent disease, who depending on their point in treatment were either therapy naive at the time of biopsy or therapy exposed. To address the lack of clarity we changed the legend in Supplementary Figure 2 and added the following sentence to the manuscript:

"The final cohort included a single sample each from 114 pediatric adolescent and young adult patients (median age = 12.6 years, range: 4.5 months to 43.8 years) with solid tumors (S.Table 1 and S.Figure 1a-d)."

Comment 8 response:

This has been satisfactorily addressed.

Comment 8:

Figure 1 also needs a title and the part b component should be color-coded to describe which types of alterations came from somatic DNA, germline DNA or tumor RNA, for example.

In the revised manuscript we have now moved panel Figure 1b to S.Figure 2. Where appropriate we provided a more detailed description of where each annotation derives from in legend instead color-coding.

Comment 9 response:

Comment 5 from reviewer 3 surrounds a different topic than that raised below. I find this more closely associated to comment 1 for which I have provided feedback above.

Comment 9:

The description of cWGTS that follows, again seems to focus on the somatic genome and provides only minimal information on how the RNA-seq is being evaluated; fusions and gene expression "signatures" (although these really aren't presented, only single gene overexpression is shown).

This concern overlaps with Comment 5 from the same reviewer and the response to Comment 5 should address Comment 9 concern.

Comment 10 response:

This has been satisfactorily addressed.

Comment 10:

In the panel concordance section on page 5, there is a reference to "prior findings" but it is unclear what findings these are and how they were derived. So provide a reference or clarify for the reader.

We thank the reviewer for the opportunity to clarify the reference to prior findings. We have now incorporated the appropriate citation in this section, which refers to mutational profiling by panel sequencing in the Zehir et al, Nature Medicine 2017 publication and changed the sentence as follows:

"Consistent with prior findings⁴ demonstrating that patients with rare cancers do not yield clinically relevant biomarkers by panel sequencing, most patients in our cohort (76%, n=87) had no therapy-informing alterations."

Comment 11 response:

This has been satisfactorily addressed.

Comment 11:

If 76% had no therapy-informing alterations from MSK-IMPACT, yet the ability of WGS to only call 81% of oncogenic mutations vs. IMPACT leads one to question whether there were more therapy-informing alterations identified by WGS?

Please state this yield. It is clear from figure 2h that the overall yield of variants was higher from WGS, but unclear whether knowing these variants mattered clinically.

The orange section (solid and mesh) from figure 4b shows the proportion of patients that had additional relevant findings by cWGTS which was 54%. The relevant additional findings are also described in Supplementary Table 2 along with clinical classification. This yield and clinical relevance are expanded upon in text:

“Consistent with recent studies 7–12,13, cWGTS analyses identified at least one additional cancer-associated oncogenic variant in 54% of patients (n=61). Of these, 33 were of direct clinical relevance including 12 diagnostic (36%), 15 prognostic (45%), 5 therapy informing (15%), and 6 germline (18%) biomarkers (Figure 4a, S. Table 2)”

Comment 12 response:

This has been satisfactorily addressed.

Comment 12:

I also found figure 2f to provide little information, so it could be eliminated.

We agree with the reviewer, that 2f over-emphasized the point of ITH which is clearly demonstrated by the results in main text, and supplementary data. In agreement with the reviewer’s suggestion, we have now eliminated Figure panel 2f.

Comment 13 response:

This has been partially addressed. While cWGTS analyses extended the detectable variant and signature spectrum, the description of the extension of findings is challenging to follow and difficult to discern from Figure 4a and Supp Table 2. This is particularly true when referencing that “less than half of the findings in cWGTS could be captured by WES and RNA-seq” (Line 176, page 8). Recommend to add to the text that the majority represented SV, and to clearly indicate in a column in Supp Table 2 which findings were detected by panel sequencing and which would have been captured by exome and RNA sequencing. Also, it is of interest for the reader to know if other well-described and clinically relevant genomic alterations were present in these exact samples which would have been captured by exome and RNA seq or if this new finding represented the only disease-associated finding of relevance. This speaks to understanding how broadly use of cWGTS extends the analysis for clinically relevant findings,

vs adds to an existing broad repertoire of detected variation in a given sample using existing well-described techniques.

Comment 13:

The labeling of figure 2g doesn't match the text description on page 7 lines 141-147 in terms of the tests used, so this is confusing. In figure 2h, while it is true that there are "novel findings" only possible from cWGTS, (really from WGS), over 50% of the novel findings are SVs, but what proportion of those were clinically meaningful vs. not?

Figure 2g, which is now figure 4a, refers to additional relevant events detected by cWGTS and color-coded by whether RNA or WES alone would also be able to pick up these events. We added more detailed description to the figure title and legend as well as this in text:

"We further inferred the portion of additional findings that would be captured by whole exome sequencing (WES) by masking results to the coding regions of genes. Of the 61 patients with incremental findings by cWGTS, RNA-seq and WES alone would only capture events in 10 (16%) and 8 (n=13%) patients respectively, or in 17 (28%) patients when combined (Figure 4a). Thus, less than half of the findings in cWGTS could be captured by WES and RNA-seq."

While there are many "novel findings" only possible from cWGTS, we are only showing relevant events in Figure 2h, which is now Figure 4b. Over 50% of the additional findings are disease relevant structural variants. We have made this clearer in text and the figure legend and provide a case-by-case description of novel findings in S. Table 2.

Comment 14 response:

This has been partially addressed. Greater clarity for the reader could be provided in adding the tumor type to figure S. Figure 5. Additionally, a novel fusion does not necessarily constitute a driver event or an alteration of clinical relevance, even if in-frame and validated. Recommend to include an additional column in S. Figure 5 if either partner has been previously described in an in-frame fusion event within cancer, and within the specific tumor type. This is important given the author's point that "this speaks to the importance of WGTS vs WGS or RNA-seq in the clinical setting."

Comment 14:

Fusions: were the novel fusions on page 9 validated? are they in-frame fusions?

We clarify that the reported fusions were in frame and that they were supported by DNA and RNA-seq data. We include the sentence:

“Eight in-frame fusion genes were identified from WGS and RNA-seq in patients with no prior findings on clinical testing (S.Figure 5, S.Table 5), 5 of which were novel.”

All fusions reported in the study were supported by DNA and RNA-seq data. This speaks to the importance of WGTS vs WGS or RNA-seq in the clinical setting. We highlight this point in the results section.

Comment 15 response:

This has been satisfactorily addressed.

Comment 15:

Additional findings: Structural Inactivation of tumor suppressors: I honestly think the most exciting findings from this study are presented in this section around the p53 alterations as illustrated in Figure 3 e. I would encourage the authors to spend more time on the ramifications of these findings, and to provide more detail on whether similar alterations could be identified in the germline of these patients. Figure 3 also needs a title, which will be difficult to come up with because it involves several different aspects of RNA and DNA-based findings.

We thank the reviewer for highlighting this important finding in our study. Indeed, in the results section and to the extent findings were supported by our data, we highlighted the importance of mapping these SVs, concordance with MSK-IMPACT, the relevance of RNA-seq integration to validate these events as functional and ramifications for correlative studies and therapy decisions. Additionally, we have clarified in the text that we did not identify germline TP53 rearrangements in our dataset by including the following sentence:

“We did not identify germline SV’s targeting the TP53 locus”.

The unifying theme in Figure 3 (which in the revised manuscript is Figure 5) is the relevance of integrating WGS and RNA-seq analysis in variant annotation and reporting with examples shown for the use of RNA-seq in resolving annotations for a) complex SV resulting in novel in- frame fusion gene, b) characterization of a non-coding SV targeting regulatory region of a cancer oncogene resulting in an enhancer hijacking event, and c) intronic SVs resulting in loss of TP53. To improve clarity in our presentation and to address the reviewers concerns we have now included a legend for Figure 3 and a thematic title “Integration of DNA and RNA findings for variant annotation”

Comment 16 response:

This has been satisfactorily addressed.

Comment 16:

For the information regarding detection of WGD, were these all cases with odd numbered WGD? i.e. triplication, etc.? Any examples of quadruplication or other even numbered events? Any validation of these by array or cytogenetics?

In agreement with prior literature linking WGD to adverse prognostic outcomes, in our analyses we classified WGD as present or not. To our knowledge the clinical relevance of resultant ploidy (i.e., triplication) is not supported by established literature and thus beyond the scope of this study, which focuses on well-established oncogenic findings. Orthogonal data to validate WGD was available for 29 of 30 neuroblastoma tumors in the study. FISH probes were used to assess the number of copies for chromosomes for 1p, 1q, 2p, 11q and 17q. In all 29 cases, predicted CN states for these chromosomes were in agreement with the FISH evaluation suggesting that the genome-wide ploidy estimates were accurate.

Comment 17 response:

This has been satisfactorily addressed.

Comment 17:

Biological and clinical implications of TMB: on line 259, another reference to 'previous reports' is mystifying and herein the first evidence to my knowledge that many of the tumors in this study are related as primary:recurrent pairs. In Figure 5A, the figure needs to better reflect which primary and treatment-resistant tumors are related to one another, and color-code the recurrent cancer with respect to the therapy used.

On original submission line 259 we provide explicit citation of the relevant references that discuss TMB as we note:

We observed higher overall (8.3-fold) TMB estimates in our cohort, 259 relative to previous reports (Figure 5a)5,39,42, where citations 5, 39 and 42 present data on TMB in pediatric tumors.

We have clarified in comments 2 and 8 that there were no tumor pairs in the study and have addressed this concern in the main text (please see response to comment 2)

Comment 18 response:

This has been satisfactorily addressed.

Comment 18:

In the description of the first case (H135022), it is difficult to understand how the finding of higher fusion and SV burden translated into the prescription of checkpoint blockade therapy, since TMB (focused on point mutations) is the only clinically approved predictor of response and this patient had only moderate TMB.

The reviewer is correct that TMB is the clinically validated predictor of response to checkpoint blockade. However, the burden of other mutational classes, specifically SV, has not been adequately evaluated since WGS and RNA-seq were not used in those studies. At the time of tumor profiling patient H135022 had progressive chemoresistant metastatic disease and had exhausted all available treatment options. The observation of high SV burden and high fusion burden was the basis for the therapeutic hypothesis that resulted in the patient being treated with ICI, absent any other available options.

In agreement with the reviewer's concern, we have now emphasized this consideration in the main text as follows:

“These findings demonstrate the value of cWGTS to fully assess the level of genomic instability across variant classes and highlight the need to further evaluate SV and fusion gene burden as biomarkers of response to immune checkpoint blockade therapies. “

Comment 19 response:

This has been satisfactorily addressed.

Comment 19:

In the second case, H135462, there is a reference to 'clinical sequencing' of primary and relapse tumors. What does this mean? Was TMB or MSI evaluated by this clinical sequencing?

By clinical sequencing the initial tumor was TMB-Intermediate (6.1). The relapse tumor was TMB-high (10.5). Both samples were microsatellite stable. Clinical PD-1 staining and tumor infiltration staining were not performed. We assessed the diagnostic and relapse sample (data for relapse not shown in the paper) by WGS and both were SV high.

What did H&E or IHC from pathology show in terms of PD-1

or other immune infiltration, etc.? Clinical PD-1 staining and tumor infiltration staining were not performed as not indicated.

The rationale for ICI for this patient was the same as for patient H135022, relapsed refractory patient at the end of therapeutic options.

Comment 20 response:

This has been satisfactorily addressed.

Comment 20:

I find the section on technical considerations and figure 6 in its entirety to be poorly presented and confusing. What aspect of feasibility or clinical utility does this section and figure actually address? It could all probably be eliminated.

In this section we address three main technical considerations with regards to WGS implementation in clinical sequencing which include: 1. Sensitivity of detection of actionable mutations typically detected by high depth sequencing; 2. Relationship between sequencing coverage and depth and 3. Variant cell fraction.

In response to the reviewer's concern, we have now moved this section further up in the manuscript to the Results section: Technical considerations: Optimal depth of coverage for clinical sequencing and Figure 3. This follows the section on tumor purity and in agreement with the reviewer's concern enables a more logical flow in the narrative. We have also simplified the corresponding figure panels and added more explicit descriptions in the main text and figure legends.

Comment 21 response:

I am in agreement with Reviewer 3 regarding the paucity of data in support of cell-free DNA studies for inclusion in this manuscript. There is particular concern surrounding the lack of reproducibility of findings, which may be attributed to low cf fraction or other technical challenges in the assay. Recommend to remove.

Comment 21:

Similarly, the study of cfDNA vs. WGTS (really WGS) of the tumor is interesting but the numbers of patients evaluated in this way are quite small and probably inconclusive.

The focus of the present manuscript is on the technical considerations of WGS in clinical oncology. We show that for some patients, suitable primary biopsies are challenging to access or limited by tumor purity. The cfDNA section demonstrates for the first time the possibility to derive full coverage WGS data and comprehensive mutation profiling to include copy number, small mutations and SV's from cfDNA. This opens the opportunity for comprehensive genome profiling for patients with limited biopsies, with high cfDNA content which is frequently the case for patients with progressive metastatic disease in need for therapeutic options. The ability to derive high quality WGS data from cfDNA has never been demonstrated before and we believe that these exploratory results represent an advancement in the field. The inclusion of these pilot results is a logical conclusion to the paper, since clinical implementation of WGS must consider the breadth of samples that may have utility in the clinic.

To address the reviewers concern we have emphasized that the cfDNA section represents an exploratory analysis, providing proof of concept data that warrant further optimization and validation in future studies in the main results section and the discussion.

Comment 22 response:

This has been partially addressed. It is unclear to what extent cWGTS actually extends clinical utility for diagnosis, prognosis, or therapeutic options. It is of interest for the reader to know if other clinically relevant genomic alterations were present which would have been captured by exome or RNA seq or if this a finding derived from cWGTS represented the sole disease-associated finding of relevance. This speaks to understanding how broadly use of cWGTS extends the analysis for clinically relevant findings, vs adds to an existing broad repertoire of detected variation in a given sample using existing well-described techniques. S.Table 2 provides presence or absence of an identified event, however much greater interpretation and extension of the clinical utility of these data would occur if all oncogenic and clinically relevant variants are documented in the table, along with designation as to which were detected solely by cWGTS.

Comment 22:

Finally, it would be interesting to understand the proportion of patients for which additional, medically meaningful findings were obtained, either in the primary or recurrent setting, that indicated new therapeutic aspects not uncovered by prior testing. This is only one aspect of clinical benefit that would meaningfully support the advocacy for cWGTS.

We thank the reviewer for this comment. We would like to clarify that in our study we organized findings as clinically relevant when they altered or guided any aspect of a patient's clinical management to include:

1. Germline predisposition findings which can inform on screening and surveillance for the patient and their families as well as treatment decisions (e.g., MMR deficient tumors for immune checkpoint blockade).
2. Biomarkers that change or refine disease diagnosis (e.g., change in diagnosis from relapsed neuroblastoma to Schwannoma).
3. Prognostic biomarkers that identify high-risk patients unlikely to respond on standard of care treatment, directing surveillance and importantly upstage of treatment per standard of care to protocols tailored towards high-risk patients.
4. Biomarkers that inform on inclusion/exclusion criteria for clinical trial entry (e.g., TP53 rearrangements).

In the results section titled "Additional findings of biological and clinical relevance detected by cWGTS" we specifically note:

We assessed additional findings of clinical relevance by cWGTS over state of the art panel sequencing to include somatic MSK-IMPACT4, MSK-fusion26 and panel testing of 88 cancer predisposition genes²⁵. Consistent with recent studies 7–12,13, cWGTS analyses identified at least one additional cancer-associated oncogenic variant in 54% of patients (n=61). Of these, 33 were of direct clinical relevance including 12 diagnostic (36%), 15 prognostic (45%), 5 therapy informing (15%), and 6 germline (18%) biomarkers (Figure 4a, S. Table 2).

And in S. Table 2 provide a case by case description of each one of these findings.

The categories described as clinically relevant are in line with emerging studies in the field (Newman et al. (2021), Wong, Mayoh, Lau et al. (2020), Horak et al. (2021)), and the results are consistent.

Key Points To Address

We would like to thank the reviewers for their careful consideration of our manuscript and their thoughtful feedback. We believe that with this constructive input the manuscript is significantly strengthened and we thank the reviewers for their insights and suggestions.

In this response letter we address a point by point response to each of the reviewer's comments.

A shared concern amongst reviewers comments was the small dataset on the cfDNA analysis. To address this shared concern we have highlighted the novel aspects of our study and delineated where our results validated established literature. We would like to thank the editors and our reviewers for considering this revised version.

Reviewer #4 (Remarks to the Author)

Review Shukla et al.

R4.Comment 1

Lines 77-80. This is largely incorrect. FFPE material and cfDNA suffer from the same low purity issue (in combination with WGS). The solutions implemented for these input materials (i.e. panel-based deep sequencing) are equally applicable to FF material. So in my opinion this sentence should convey the message that more sequencing depth is required for lower purity samples, which is currently not economical, but with decreasing sequencing prices may become feasible. Alternatively, (laser-capture) tumor enrichment and/or panel-based approaches would address the challenge. I don't think FFPE or cfDNA do make sense to bring up here as a solution for low purity issues.

Response: We agree with the reviewer that FFPE and cfDNA suffer from the same low purity issues. In the original text we addressed two concepts in the same sentence: 1. purity and 2. sample source availability. We appreciate that this led to reduced clarity of the message.

Prompted by the reviewers comment we reviewed the low purity attrition samples. Of the 201 nominated samples, 58 were enriched for post therapy biopsies (43/58) and failed path review on the basis of tissue necrosis, calcification etc. These samples were not submitted to sequencing. Of the tumors that were submitted for sequencing only 29 failed on tumor purity as assessed by WGS. This

highlighted an unappreciated effect of treatment on tumor viability for NGS sequencing. We have thus revised the relevant results section as follows:

The study cohort included patients presenting with primary diagnostic or relapse/refractory disease. Of 201 patient fresh frozen (FF) tumors nominated for paired cancer/normal whole genome and transcriptome sequencing (cWGTS), 58 were excluded upon pathology review and 29 did not meet our requirement for >20% tumor purity as assessed by WGS. The majority of the excluded cases were post-therapy neuroblastoma and sarcoma samples with a predominance of necrotic disease.

R4.Comment 2

Line 92/93. This is not correct. See [https://www.jmdjournal.org/article/S1525-1578\(21\)00120-3/](https://www.jmdjournal.org/article/S1525-1578(21)00120-3/) reporting 10 days. In general, proper reference to and comparison with the results in this paper are largely lacking the current manuscript.

Response: The reviewer highlights the paper by Roepman et al which was published while our paper was in review, hence the omission. We would also like to clarify in Fig. 1b of our original manuscript submission that we compared the best reported turn around time frames to remain consistent with the TAT reported for all comparator studies. To address the reviewer's concern we have now included the results from Roepman et al in our comparison figure (Fig. 1b).

Another important point is that the Roepman timeframes represent TATs of scale as the Hartwig Medical Foundation has established a national WGS facility, including robust sampling procedure and logistics in >45 (of the 87) hospitals located across the Netherlands for the centralized analysis of tumor biopsies by WGS. In our manuscript we demonstrate that clinically relevant timeframes can be achieved within the diagnostic workflows of a single institution.

We have now included the following statement to highlight this important aspect:

This is shorter than the standard turnaround time (TAT) for many clinical NGS panel sequencing tests (2-4 weeks)^{1,2,4} markedly faster than the majority of WGS processing time frames in literature (3-8 weeks, Fig. 1b)¹¹⁻¹⁶ and comparable to the TAT achieved by centralized infrastructures of scale such as the Hartwig Foundation¹⁸.

R4.Comment 3

It is also not clear if for all 114 patients WTS data was successfully generated (and with what depth and QC metrics). Statistics on the success rates for this should be added to the sample processing section.

Response: We thank the reviewer for highlighting this omission. To address this concern we have now included Supplementary Table 2 with detailed QC metrics generated for both WGS and RNA-seq. Additionally we provide:

- Supplementary Fig. 1c: Representation of assays generated for each sample.
- Supplementary Fig. 1d: Distribution of WGS coverage in the sample set.
- Methods section: Detailed information on QC checks performed on WGS and WTS.

R4.Comment 4

Figure 1B is misleading. Best reported turnaround time is not very informative from a clinical perspective. This graph should be converted in median reported turnaround time (preferably with ranges). Own data about the four consecutive batches (line 89) should all be included in this figure to highlight the speed and impact of learning curve and optimization.

Response: We thank the reviewer for pointing to this. To the reviewer's concern:

- We used best TAT to be consistent with the comparator literature as ranges have not been provided for most studies.
- Best reported turnaround timeframe in our study reflects the currently optimized and locked down workflow for sample processing.

To address the reviewer's concern we now provide the range of our TAT as well as best reported TAT in Fig. 1b and in the main text as follows:

Time logs were audited starting at the time of surgical biopsy submission to report delivery for review by an interdisciplinary molecular tumor board for samples in our study with audit trails recorded through our biosciences platform (Supplementary Fig. 1e). End-to-end, this workflow was executed on average in 17 days during the developmental phase (range: 11-29, n=59 samples), reaching a fully optimized workflow with a final turn around of 9 days (Fig. 1b, n=16).

Last, we provide end-to-end TAT audit trails for 59 samples, excluding the 16 cases that represent our final optimized TAT. Please note that owing to a transition into our new version of our bioscience platform we only have TAT for these 59 + 16 samples. To address the reviewers concern we further provide a detailed breakdown of the TAT and the respective components in Supplementary Fig. 1e.

R4.Comment 5

More than 20% of variants are discordant between MSK-IMPACT and WGS and are explained by ITH. This number is quite high. Is this because primary tumor samples were analyzed, which are known to be more heterogeneous than metastases? Were any metastases included and was the fraction of discordant observations similar? The main body of the text does not include information on the source of the material analyzed and stage of the disease. It would be useful to add such information to be able to better interpret the ITH numbers.

Response: Our discrepancy rate (20%) is similar to what has been observed in comparison of matched WES/WGS analyses in 746 TCGA samples, where 80% of the mutations in exonic regions was detected in both platforms [PMID: 32958763].

ITH is a well established source of discordant calls when comparisons are made from DNA derived from different biopsies. Furthermore, in our study we validate ITH as the cause of the discordant calls by repeating MSK-IMPACT sequencing on the FF derived DNA (same DNA aliquot used to generate WGS libraries) which shows 100% concordance.

With regards to potential biases that may originate from primary vs metastatic tumors. Tumors from the primary sites, metastases and local recurrences constituted 47%, 38% and 15% of our cohort. We have modified Supplementary Fig. 1b to include a barplot to summarize this information. Discordant calls were observed in 30%, 46% and 24% primary, metastasis and local recurrences suggesting that there is no significant over-representation of discordant calls across these sample types.

However our cohort is fairly small and not suited to systematically address this question. Larger datasets such as the Hartwig Medical Foundation cohort [PMID: 31645765] are better suited to draw definitive conclusions between tumor stage, ITH and concordance in calls that the reviewer is suggesting.

R4.Comment 6

The statement in line 135 (cWGTS had 100% concordance) is misleading. This is not proven because only 44 out of 47 discordant variants were followed up. You can only state: at least 99% or something.

Response: To address the reviewers concern we have now edited the relevant sentence to:

These validation studies demonstrated that discordant calls were due to stochastic sampling of heterogeneous tumors²⁶ (Supplementary Table 4) and concordance between WGS and MSK-IMPACT is at least 94% (43/47 total discordant) and up to 100% for all 43 mutations in evaluable samples when the same DNA aliquot was used in both assays.

This statement is misleading in second way as the comparison should always be done in both directions to conclude something about concordance. How many variants were detected by WGS in the target regions of MSK-IMPACT that were not seen/called by MSK-IMPACT (potential false positives by WGS or false negatives by MSK-IMPACT). This analysis should be added to the manuscript to make concordance claims solid.

Response: We would like to confirm that the two way comparison proposed by the reviewer was already included in the main text of the original manuscript submission as follows:

Further corroborating ITH, WGS and targeted re-sequencing data of the same FF DNA aliquots identified 10 mutations (VAF: 6-31%), which were not reported by MSK-IMPACT from the patient-matched FFPE

*sample. Three of the 10 additional calls were cancer-associated variants (TP53 L265Yfs*81, PPM1D S468*, and HLA-A L102Hfs*73)* (Supplementary Fig. 3e).

And as a minor comment, cWGTS should read WGS in this sentence as no RNA data was used here.

We agree and have updated the manuscript to consistently use WGS or RNA-seq when we compare findings to either assay, but maintain the use WGTS when we include results from both WGS and RNA-seq.

R4.Comment 7

The added value of WTS is poorly described. Did WTS yield more relevant biomarkers than the 18 fusions? As these were all also detected by WGS, shouldn't you conclude that the added value of WTS is rather limited from a diagnostic perspective? When the WGS findings are solid and validated, there is formally no need for an independent orthogonal validation.

Response: On broader clinical utility of RNA-seq, as the reviewer says, RNA-based profiling is not as well established as DNA-based profiling. However, the utility of RNA-seq extends beyond identification or validation of fusion genes that can readily be detected by WGS to include interpretation and prioritization of SVs that may result in oncogenic and clinically relevant mutations. We highlight this under-appreciated point in the results section **“Integration of RNA-seq and WGS for variant annotation”** where we present diverse case studies that demonstrate the added value of RNA-seq in the determination of clinically relevant biomarkers to include:

1. Detection and validation of fusion genes (Fig. 2f and Supplementary Fig. 5).
2. Detection and validation of fusion genes or enhancer hijacking events generated by complex genomic rearrangements such as chromothripsis or chromoplexy that by SV detection alone cannot definitely determine the resultant event. Examples include:
 - a. A chromoplexy leading to over-expression of MYB via hijacking of NFIB enhancer in H133676 (Fig. 5a).
 - b. A chromothripsis involving 29 SVs and resulting in PAX3-FOXO3 fusion in H134768 shown in Fig. 5c.
 - c. A chromothripsis involving 12 SVs and creating MAN1A2-ACBD6 fusion in H134756 (Supplementary Table 9).
3. Detection and validation of non-coding structural variants that result in loss of function of key cancer genes (e.g TP53 intronic SVs shown in Fig. 5e) .

Simple SV's resulting in established fusion genes (category 1) can be identified by either WGS or RNA-seq. However, simple SVs that result in fusion genes are not the only clinical relevant events. In our manuscript we demonstrate specific examples within categories 2-3, where SV calling by WGS alone did not deliver definitive evidence for consideration in a diagnostic setting and where the supportive

evidence from RNA-seq was pivotal in validating and prioritizing these SV events as clinically relevant findings.

The section on integration of RNA-seq and WGS is largely descriptive and is insufficiently quantitative to judge the added value of RNA-seq for diagnostic testing. I would like to note that I do not want to deny the value of WTS in general, but it's utility in the clinic is currently largely anecdotal and not coupled to formal actionability as far as I am aware.

Response: To further substantiate our integrative analysis of RNA-seq and WGS, in this version of the manuscript, in response to R5 comments, we perform for the first time a systematic integration of aberrantly expressed genes with SV findings from WGS. In Supplementary Table 3 we provide a breakdown of the expression biomarkers into those that 1) are tissue-specific 2) overlap into regions rearranged by structural variants or 3) neither. This analysis shows that in 29% of patients (29/101), at least 1 gene with tissue-specific expression was over-expressed in a tumor originating from that tissue and in 55/101 patients, at least 1 gene falling within a genomic region rearranged by structural variants had an aberrant expression. To this end, we have changed Fig. 4, Supplementary Fig. 2, Supplementary Fig. 4. We also updated the results section to include:

Global gene expression signatures were further used to cluster samples by tumor type, providing further opportunity to resolve a patient's diagnosis (Supplementary Fig. 6a). Last, in the 101 patients with RNA-seq data we identified on average 18 gene expression biomarkers per sample (range=1-99)¹⁶ (Supplementary Table 10). However, the evidence with regards to the clinical utility of such expression biomarkers remains to be validated. To this end, we performed a systematic interrogation of genes with aberrant expression with known tissue-specific expression patterns and SVs or fusions detected from cWGTS. Overall only 8% of the expression biomarkers were associated with an acquired SV or fusion gene, demarcating a subset of high confidence expression biomarkers (average per patient = 1, range= 0-10). This limits the number of patients in our cohort with an expression biomarker supported by an SV to 54% (n=55) (Supplementary Fig. 6b). Demonstrating the utility of this integrative analysis, we identified a concomitant KRAS amplification and overexpression in two patients with no clinically relevant biomarkers (H135462 and H195916) by clinical testing. Of note in both patients the amplification was not reported by MSK-IMPACT pointing to the lower sensitivity to detect copy number changes by panel-based assays.

I think this section would benefit from a clear split (table?) in numbers of cases where WTS alone provided actionable information, where WTS supported the WGS results and where a conclusive outcome was only possible based on WGS and WTS. This quantification is highly relevant as choosing for WGS or WGTS also needs a balance between added value and added costs, which is not yet easily possible with the data currently provided.

Response: To address this concern we have now included a new column titled "Platform that detects" to Supplementary Table 3 annotating if an additional finding can be detected by cWGTS only or if by

nature of the event whether WES in combination with RNAseq would have captured the event. 17/62 of the additional findings would be detected by an integrated analysis of WES and RNA-seq. Additionally, we modified the corresponding results section to include the following sentence:

“only 27% of the new findings in cWGTS could be captured by WES and RNA-seq as majority of additional findings were attributed to SVs.”

R4.Comment 8

The additional structural variants targeting tumor suppressor genes could benefit from some more details on the nature of the SVs. Are these all (small) deletions, or just gene-disrupting break-junctions, or part of complex events, etc. This helps understanding why these are additional WGS findings

Response: To address the reviewers concern we have now included detailed information on the type of oncogenic SV's in Supplementary Table 9. These variants are not detected by targeted panels because they predominantly result from complex events such as chromoplexy and chromothripsis that highly rearrange the affected loci (65% of the events). Interestingly, most of the remaining events were simple deletions (26%) with a median length of 181Kb (range=20Kb-29Mb) confirming the established notion that targeted panels are not suitable for sensitive detection of structural variants and copy number aberrations.

R4.Comment 9

WGD rates are much lower than reported in <https://www.nature.com/articles/s41586-019-1689-y>. Is this due to differences in tumor type distribution or does this relate to primary versus metastatic lesions?

Response: WGD rates depend on histology and disease stage [PMID: 24071852, 31645765]. We reported WGD in 40/114 patients (28%). The rate in our study is comparable to what has been reported for primary tumors from adult onset cancers (25-37% in PMID: 24071852 and 22544022) but this would need further validation in a larger pediatric cohort.

The referenced dataset by Priestley et al [PMID: 31645765] is not an appropriate comparator for our cohort. 62% of our cohort are from primary/local-regional tumors while Priestley et al studied metastatic tumors from adult onset cancers, predominantly carcinomas. As suggested by Priestley et al, metastatic tumors might have a higher prevalence of WGD events. The proportion of carcinomas is very low in our cohort. In our manuscript the emphasis is that WGD represents an additional genomic biomarker that can be detected by WGS, and for the reasons outlined above we do not foray into any claims on incidence etc.

R4.Comment 9

TMB estimates were overall 8.3 fold higher than previous reports. This needs some explanation. I guess previous studies were panel of WES-based. I would be good to also report on the coding TMB as it is known that mutation rates differ between coding and non-coding sequence. Could this at least partially explain the difference?

Response: We would like to clarify that in our original manuscript we already provided estimates on coding TMB (Figure 7a, Supplementary Figures 4b and 8a).

Importantly, the TMB estimates in our study are comparable to the largest pediatric WGS study from Zero Childhood Cancer Initiative [PMID: 34112699] 1) with a similar composition of tumor samples: 47% diagnostic and 53% relapse/refractory disease vs 48% diagnostic and 52% relapse/refractory in our cohort and 2) comparable depth of sequencing: median 90x vs 95x in Zero Childhood and our cohort, respectively.

The discrepancy with Grobner et al is likely owing to differences in median WGS coverage where Grobner et al used ~30x coverage compared to ~95x in our study. We used Grobner et al as a comparator as at the time of manuscript submission this was the only appropriate reference dataset. To address the reviewers comments we have now included the Zero Childhood Cancer Initiative data for the comparator dataset in Fig. 7a.

R4.Comment 10

The cfDNA analyses on only seven patients is insufficiently powered to be informative and does not add to the main message of the manuscript or the data already present in scientific literature (e.g. <https://www.nature.com/articles/s41467-021-23445-w>, which provides much more solid numbers). I believe this part would better be removed from the manuscript.

Response:

Per the reviewers suggestion we fully acknowledge that the cfDNA results section is exploratory and provides proof of concept data of the feasibility to derive comprehensive WGS data from cfDNA data. To this end, we highlight the exploratory nature of our analysis and limited sample size in the main text as follow:

In an exploratory analysis, we performed WGS from matched FF samples and cfDNA from seven patients collected tumor biopsy (Supplementary Table 11).

and

These results provide the first proof-of-concept for the feasibility of deriving tumor agnostic comprehensive WGS profiling from a liquid biopsy.

In the discussion we further highlight that these results serve as proof of concept and the need to conduct future validation studies exploring in detail the utility of cfDNA WGS in clinical oncology by incorporating the following text:

We provide proof-of-concept feasibility data demonstrating that comprehensive WGS profiling can also be leveraged in patients who have high cfDNA content in circulation at time of diagnosis or relapse. Our findings pave the way for future studies focused on analytical validation and optimizations of comprehensive tumor agnostic WGS profiling from cfDNA for diagnostic purposes.

We also change the text in the abstract as follows:

Last, we demonstrate the potential of cfDNA for tissue-agnostic WGS profiling.

In response to the reviewer's concerns on lack of novelty we would like to clarify the following points:

1. The first study mentioned by the reviewer uses low-pass WGS data (12x) to perform an integrated analysis of copy number profiles and fragmentation patterns only. The authors do not identify somatic substitutions, indels or SVs like we do in our manuscript.
2. Contrary to the reviewer's suggestion, PMID: 31563277 recently evaluated 50 different studies from literature focusing on ctDNA profiling in pediatric tumors. The majority of these were confined to the analysis of genome-wide CNA profiles (18 studies), targeted sequencing of point mutations/indels (17 studies) or patient-specific translocations/rearrangements with a priori knowledge (13 studies).
3. Our study, to our knowledge, is the first to demonstrate proof of principle that full coverage tumor WGS analysis to include copy number analysis, calling of small mutations, SVs, as well as derivation of mutation signatures and putative phylogenetic structures is possible from cfDNA. We acknowledge that this is a proof of concept analysis in the main text and discussion.

To this end, we include in the cfDNA results section:

Recently cfDNA NGS profiling including WGS has been used to detect tumor-specific CNAs and fragmentation patterns in pediatric tumors⁵⁴⁻⁵⁶. However, the potential of high-depth WGS in tissue-naive identification of the genetic changes (SNVs, indels, SVs) in a patient's cancer genome from cfDNA has been largely unexplored.

With regards to the value of the cfDNA results section suitability for inclusion in the manuscript. The focus of the manuscript is on the utility of WGS in clinical oncology. We show that for some patients, suitable primary biopsies are challenging to access or limited by tumor purity. The cfDNA section demonstrates for the first time the possibility to derive full coverage WGS data and comprehensive mutation profiling to include copy number, small mutations and SV's from cfDNA. This opens the opportunity for comprehensive genome profiling for patients with limited biopsies, with high cfDNA

content which is frequently the case for patients with progressive metastatic disease in need for therapeutic options. Thus, this is an important expansion in sample sources that should add to the clinical applicability of WGS in the future. This thematic link and novelty of the analysis should justify retention of the cfDNA section in the manuscript. We fully acknowledge that definitive follow up studies on the analytical validation of cfDNA are warranted that extend beyond the scope of this paper.

The reported cfDNA percentage is 10 to 83%. This appears extremely high compared to numbers that are typically reported in literature (e.g. median of only 0.18% in this study: <https://www.nature.com/articles/nm.1789>). This needs additional details on how these percentages were achieved and determined and a note that these purities are very atypical (and therefore the inclusion of a statement on cfDNA analysis feasibility in the abstract is not justified). What type of tumors had these patients?

Line 320: the lower limit of ctDNA should be mentioned explicitly.

All samples in the cfDNA pilot were post therapy relapses. Information on tumor type, disease stage were already provided in Supplementary Table 1 (original submission). We now also provide this information for the subset of patients that underwent cfDNA analysis in Supplementary Table 11. We confirm that no pre-screening or selection of the pilot cases was conducted on the basis of cfDNA content but was predominantly based on cfDNA sample availability. Notably, the tumor purity of the FF samples for these patients was high (median 93% range 22%-98%) which allowed us to conduct a detailed comparison between FF and cfDNA. cfDNA purity was determined using established algorithms for estimating tumor purity and ploidy. Please see the relevant methods section titled “cfDNA tumor content and variant comparison.”

Last, determination of lower limit of detection for cfDNA WGS approaches would warrant further validation studies as highlighted in the discussion.

R4.Comment 11

A statement on data availability is lacking.

Response: We confirm that all data will be available through dbGAP (phs002620.v1.p1) and cBioPortal with publication and confirm this in a data availability section in the methods.

Related to the other reviewer comments

R1.Comment 1

The procedures detailed here for clinical decision making and tumor board support should be included in the manuscript.

Response: We thank the reviewer for the opportunity to improve the manuscript in this regard. To describe our review process we have now included the following texts in the main text:

The clinical relevance of WGTS findings that were not identified by clinical panel sequencing (MSK-IMPACT⁴, MSK-fusion²⁸ and panel testing of 88 cancer predisposition genes²⁷) was determined by a multidisciplinary molecular tumor board.

and in the methods section:

Incremental findings were further classified as clinically relevant if they met one of the following criteria: 1) diagnostic finding - defined as an alteration which provided justification or an alteration in cancer diagnosis or cancer subtype diagnosis, 2) established prognostic finding - defined as an alteration with established prognostic relevance with robust support from scientific literature, 3) likely pathogenic or known pathogenic germline predisposition event, 4) treatment informing finding - defined as an alteration which provides direct justification of a therapeutic modality, or 5) a novel driver oncogenic fusion.

R1.Comment 3

The statement on ‘the ability to derive high quality WGS from cfDNA has never been demonstrated...’ is not true: see <https://pubmed.ncbi.nlm.nih.gov/30617129/> or <https://www.nature.com/articles/s41467-021-23445-w>

Response: We thank the reviewer for the opportunity to improve the clarity of this sentence. Similar to our response to R4 Comment 10, whilst there have been analyses of low pass WGS on cfDNA for copy number analysis and targeted evaluation of mutations, our study is, to our knowledge, the first to conduct de novo variant calling across all classes of mutations from high-depth WGS data derived from cfDNA. To address this concern we have included the following sentence in the cfDNA results section:

Recently cfDNA NGS profiling including WGS has been used to detect tumor-specific CNAs and fragmentation patterns in pediatric tumors^{54–56}. However, the potential of high-depth WGS in tissue-naive identification of the genetic changes (SNVs, indels, SVs) in a patient’s cancer genome from cfDNA has been largely unexplored.

R3.Comment 2

This is a main concern that I share and which is not properly addressed in the current version of the manuscript.

Response: In our manuscript we demonstrate the clinical utility of cWGTS in detecting clinically relevant events. The aim of our cfDNA analysis was to evaluate if cfDNA presents an additional bioanalyte to deploy WGS for diagnostic purposes and in our pilot we evaluated the feasibility of tumor-naive mutation/SV calling genome-wide is feasible. We argue that the novelty of our analysis presents a significant advance in the field. We fully acknowledge that definitive follow up studies on the analytical

validation of cfDNA are warranted that extend beyond the scope of this paper and include in the following sentences to clarify this need.

In cfDNA results section:

Recently cfDNA NGS profiling including WGS has been used to detect tumor-specific CNAs and fragmentation patterns in pediatric tumors⁵⁴⁻⁵⁶. However, the potential of high-depth WGS in tissue-naive identification of the genetic changes (SNVs, indels, SVs) in a patient's cancer genome from cfDNA has been largely unexplored.

and

These results provide the first proof-of-concept for the feasibility of deriving tumor agnostic comprehensive WGS profiling from a liquid biopsy.

In discussion:

We provide proof of concept feasibility data demonstrating that comprehensive WGS profiling can also be leveraged in patients who have high cfDNA content in circulation at time of diagnosis or relapse. Our findings pave the way for future studies focused on analytical validation and optimizations of comprehensive tumor agnostic WGS profiling from cfDNA for diagnostic purposes.

We also change the text in the abstract as follows:

Last, we demonstrate the potential of cfDNA for tissue-agnostic WGS profiling.

R3.Comment 6

R3, comment 6: the reviewer asks for TAT overview of all samples over time. This is still not provided in the revised figure.

Response: This has now been addressed in our response to R4 Comment 4.

R3.Comment 5 and 9

R3, comment 5 and 9: I share these concerns regarding the demonstrated added value of WTS. This is not sufficiently addressed in the current version of the manuscript. See also my comments above.

Response: We address this concern in R4 Comment 7.

R3.Comment 15

R3, comment 15: the involvement of structural variants (e.g. LOH) resulting in almost 100% biallelic inactivation of TP53 rates

Response: We are not sure as to what the reviewer means here as the sentence is not finished or was corrupted during the transmission.

Reviewer #5 (Remarks to the Author)

Expert in pediatric cancer genomics

The manuscript prepared by Shukla and colleagues describes the development and implementation of paired genome sequencing and transcriptome sequencing in prospective institutional cohort of adolescent young adult and adult patients <40y with newly diagnosed, treatment refractory or relapsed cancer. A major focus of the manuscript is the feasibility of translation from a research environment into a clinical setting.

The concepts presented herein are not novel in the sense that cancer-based exome, genome and transcriptome sequencing have been previously well-described in the literature, including validation of these methods in a clinical setting. The authors give appropriate reference to those prior studies. Areas highlighted in the manuscript include not only the expected SNV, CNV and SV/fusion detection performance, but also extend into the extraction of mutational signatures, microsatellite instability, chromothripsis, and telomere length, among others. Furthermore, the manuscript includes discussion of quality metrics for assay performance and the use of the applied bioinformatic tools in this space. Commentary and data surrounding optimal depth for genome sequencing is of great value to the reader and well described.

Overall, given the clinical focus, the manuscript would benefit from providing greater detail in the methodologic approach to genomic results interpretation and reporting. A major emphasis of the manuscript is the ability to turn around genome and transcriptomic results in 9 days. However, the very limited sample set (n=16) and time period in which this was employed speaks more to a theoretical timeline rather than a routine and established workflow. Cell-free DNA is conceptualized toward the end of the manuscript, however the very limited study (n=7), and the variability in sample preparation makes it difficult to draw clear conclusions from these data.

R5.Comment 1

1. Limited detail is provided on how variants were classified into the stated categories below and what attributes related to a variant were considered. Please expand on the decision-making process and definition of the categories below. Also, can the authors comment on why the AMP/ASCO/CAP guidelines for somatic variant interpretation not employed?

1. Diagnostic 2. Risk predisposition for germline variants, 3. Prognostic, 4. Therapy informing, 5. Pathogenic, 6. Likely Pathogenic or 7. Variant of unknown significance (VUS).

We thank the reviewer for the opportunity to clarify this point. Per the reviewers suggestion we classified variants using the aforementioned categories 1-7 also included in Supplementary Table 3.

Given that we explore findings from WGS and RNA-seq our annotation was limited to categories 1-6. Variants of unknown significance were not explicitly annotated.

For somatic variants these annotations were derived from OncoKb, the first and only FDA approved database for variant annotation. For germline variant annotation we used ACMG/ClinGen annotations as detailed in our methods section.

R5.Comment 2

While cWGTS analyses extended the detectable variant and signature spectrum, the description of the extension of findings is challenging to follow and difficult to discern from Figure 4a and Supp Table. This is particularly true when referencing that “less than half of the findings in cWGTS could be captured by WES and RNA-seq” (Line 176, page 8). Recommend to add to the text that the majority represented SV, and to clearly indicate in a column in Supp Table 2 which findings were detected by panel sequencing and which would have been captured by exome and RNA sequencing.

Response: We thank the reviewer for the opportunity to increase clarity in our results description. To address this concern we have now:

- Included a new column titled “Platform that detects” to Supplementary Table 3 annotating if an additional finding can be detected by cWGTS only or if by nature of the event whether WES in combination with RNAseq would have captured the event. 17/62 of the additional findings would be detected by an integrated analysis of WES and RNA-seq.
- We modified the text from “less than half...” to

“27% of the new findings in cWGTS could be captured by WES and RNA-seq as majority of additional findings were attributed to SVs.”

Also, it is of interest for the reader to know if other well-described and clinically relevant genomic alterations were present in these exact samples which would have been captured by exome and RNA seq or if this new finding represented the only disease-associated finding of relevance. This speaks to understanding how broadly use of cWGTS extends the analysis for clinically relevant findings, vs adds to an existing broad repertoire of detected variation in a given sample using existing well-described techniques.

Response: As requested by the reviewer, in this version of the manuscript we have updated Supplementary Table 3 to include mutations, CNAs and SVs reported by clinical testing (MSK-IMPACT, MSK-Fusion). Additionally we have introduced a column to highlight cases where clinical testing failed to identify a clinical biomarker and already include columns to identify cases with additional biomarkers by WGTS. In this column (Supplementary Table 3, Column “Any previous clinical finding”) we have marked the following categories:

- 0 - No prior finding by clinical diagnostic workflow and no additional by cWGTS
- 1 - No prior finding by clinical diagnostic workflow and additional by cWGTS
- 2 - Prior finding by clinical diagnostic workflow, no additional finding by cWGTS
- 3 - Prior finding by clinical diagnostic workflow and additional finding by cWGTS

We would like to highlight that the clinical utility of cWGTS should consider all additional findings rather than consider cases without prior biomarkers by clinical testing.

R5.Comment 3

Clear emphasis in the manuscript is focused on a 9 day TAT for cWGTS. Given the limited sample set (n=16) and restriction to only 4 batches, the time of the established workflow should be de-emphasized. Recommend to report the TAT from across the entirety of the study which seems to be more in fitting with allowing a direct comparison to other published studies.

Response: We thank the reviewer for pointing to this comment. We would like to clarify that the 16 sample pilot represents our final optimized locked down workflow, which is now routinely established in our center. Owing to a transition into our new version of our bioscience platform we only have TAT audit trails for 59 samples, excluding the 16 cases that represent our final optimized TAT. To address the reviewer's concern we now provide details on TAT for the 59 samples as well as the optimized cohort in the main results text:

Time logs were audited starting at the time of surgical biopsy submission to report delivery for review by an interdisciplinary molecular tumor board for samples in our study with audit trails recorded through our biosciences platform (Supplementary Fig. 1e). End-to-end, this workflow was executed on average in 17 days during the developmental phase (range: 11-29, n=59 samples), reaching a fully optimized workflow with a final turn around of 9 days (Fig. 1b, n=16). This is shorter than the standard turnaround time (TAT) for many clinical NGS panel sequencing tests (2-4 weeks)^{1,2,4} markedly faster than the majority of WGS processing time frames in literature (3-8 weeks, Fig. 1b)¹¹⁻¹⁶ and comparable to the TAT achieved by centralized infrastructures of scale such as the Hartwig Foundation¹⁸.

Last, unique to our study we provide a detailed breakdown of the TAT and the respective components in Supplementary Fig. 1e.

Additionally, a critical missing component is the time spent on review/ interpretation of genomic data with the interdisciplinary team who is establishing relevance in clinical practice and the frequency of such meetings.

The report produced represents the final output of the analysis, which is reviewed by an interdisciplinary tumor board that takes place biweekly. Typically 8-10 cases are reviewed in one hour. We have added the following sentence in the methods section:

Putative findings of clinical relevance identified by WGS and RNA-seq were reviewed by an interdisciplinary team of clinical oncologists, molecular pathologists and cancer genomics experts. Typically 8-10 cases were reviewed in an hour-long tumor board meeting that was held biweekly.

R5.Comment 4

- a. **Could the authors comment on employed quality metrics other than depth and tumor content and how those were determined?**
- b. **Is a QC report generated for each patient, and if so which data are encompassed in that review?**

Response (to a and b): As requested by the reviewer, we have added Supplementary Table 2 highlighting metrics employed from our patient-level QC analysis. In addition to the concordance analysis that we already mentioned in the original manuscript, we confirm that there is a QC report generated for each patient for both DNA and RNA samples and have added the following to the methods in order to detail what is encompassed:

Additionally, a quality control report is generated per sample using MultiQC (v1.9) (<https://github.com/ewels/MultiQC>) to aggregate alignment and read statistics from FastQC (v0.11.5) (<https://github.com/s-andrews/FastQC>), Picard (v2.25.6) (<https://github.com/broadinstitute/picard>) and RNA-SeQC (v1.1.8.1) (<https://software.broadinstitute.org/cancer/cga/rna-seq>)¹⁸⁻²⁰.

- c. **Additionally, can the authors provide data on the reduction in spurious variant calls through use of CaVEMan as related to the ability to filter out sequencing artifacts.**
- d. **Finally, can the authors comment on how a filtration strategy was applied in the panel of 100 unmatched normals (are these matched by tissue type- blood vs FFPE vs FF), and how many artifacts were eliminated via that approach? Understanding the initial annotated variant list followed by filtration/prioritization, and at the end, the range of variants considered in the context of clinical review would aid the reader.**

Response (to c and d): Across our cohort we see a median 3.8% [2.0%-28%] reduction in spurious variant calls through CaVEMan filtering. We employ the VUM flag from CaVEMan post-processing described here: <https://github.com/cancerit/cgpCaVEManPostProcessing/wiki/flags-and-settings>. Using a panel of 100 unmatched normals from blood, we further flag any variant with ≥ 3 mutant alleles in at least 1% of samples within the panel.

Our unmatched normal data is generated from sequencing of blood samples. Per the reviewer's comment we have further expand on VUM metrics in the method sections:

Variant post-processing was done using default flags for Strelka2 and MuTect2 while for CaVEMan, cgpCavemanPostprocessing (v1.5.2) was used filtering for sequencing artifacts with ≥ 3 mutant alleles in

at least 1% of samples within a panel of 100 unmatched blood normals (<https://github.com/cancerit/cgpCaVEManPostProcessing>).

R5.Comment 5

Define the term “oncogenic” which is used throughout (e.g. page 6, line 132; Figure 4a)

Response: We have now included the following wording in the methods section:

We define a variant as oncogenic if it represents an established “driver” mutation on the basis of prior literature and recurrence in cancer genome. For SNVs, indels and fusion genes these annotations are derived from OncoKb. For SVs we annotate events that target known oncogenes and tumor suppressor genes and use prior literature as reference (e.g for TERT, ATRX and TP53 SVs).

R5.Comment 6

6. In reference to a germline alteration, refrain from using the term mutation, as variant has become the standard term in the field. Please update throughout. Additionally, in the methods section under the heading “Identification of Germline Mutations”, the term SNP is used. Please update to SNV as germline disease-associated variants are not polymorphisms.

We thank the reviewer for this suggestion and confirm that we now use “variant” when referring to germline alterations.

R5.Comment 7

7. A drawback of this study is limitation to fresh, frozen tissue with a significant level of attrition due to failure to meet tumor purity requirements. Can the authors comment on the type of biopsies (core needle vs resection, etc) and the typically provided size/weight estimates of the available tissue to provide the reader greater clarity on the nature of the source under study.

In this study, the vast majority of samples were obtained by tumor resection or excisional biopsies (as opposed to core needle biopsies). Of the 114 cases; only 3 were obtained by core needle biopsy. The remaining 111 were obtained by excisional biopsy or tumor resection.

The high attrition rate was primarily due to post-therapy samples. All patients who were undergoing biopsy or resection of a solid tumor were eligible for this study regardless of past therapy. The majority of tumors which failed purity requirements were post-chemotherapy sarcomas and neuroblastomas with subsequent treatment-related necrosis.

To address the reviewers concern we have revised the main text as follows:

Of 201 patient fresh frozen (FF) tumors nominated for paired cancer/normal whole genome and transcriptome sequencing (cWGTS), 58 were excluded upon pathology review and 29 did not meet our

requirement for >20% tumor purity as assessed by WGS. The majority of the excluded cases were post-therapy neuroblastoma and sarcoma samples with a predominance of necrotic disease. The final cohort included a single sample each from 114 pediatric, adolescent and young adult patients (median age = 12.6 years, range: 4.5 months to 43.8 years) with solid tumors (Supplementary Tables 1-2, Supplementary Fig. 1a-d).

R5.Comment 8

8. Were orthogonal methods applied to validate the signatures data sets generated from cWGTS? For instance, how was accuracy in telomere length assessed? This is of interest given the very modest association with length relative to mutational status (Supp. Figure 7).

Response: We used TelSeq [PMID: 24609383] to estimate telomere content from WGS. The authors of the original paper performed orthogonal validation comparing the NGS-based telomere length to Southern blot measurements terminal restriction fragment lengths. Thus we did not further perform additional orthogonal validation for telomere length assessment as this is beyond the scope of our study.

In the revised version of the manuscript we now include statistical comparisons and demonstrate that the associations presented in Supplementary Fig. 7b-c are statistically significant. Moreover the associations described between telomere length and ATRX events are consistent with what has been reported in the literature [PMID: 33627664, 32024817]. We include these papers as citations in the main manuscript.

R5.Comment 9

Please provide greater detail on the calculation employed for TMB and the definitions of hypermutator and pediatric high referenced in the supplemental.

Response: We have now included a section in the methods titled "Calculation of tumor mutational burden" which reads:

Tumor mutational burden (TMB) was calculated using high confidence, somatic substitutions and indels that fall within coding regions. The totals for these variant classes were combined and then converted to coding TMB using a divisor of 30 to approximate the length of the human exome in Mb. Values greater than 2 coding mutations per Mb were considered pediatric high and values greater than 10 coding mutations per Mb were considered hypermutators, thresholds set by the study in Grobner et al³⁸.

R5.Comment 10

10. Among the variants unique to MSK-IMPACT, was manual review of the panel and genome data performed to discern if there was an underlying contributor such as strand bias, alignment issues,

base quality, repetitive sequence, or other cause as opposed to intratumor heterogeneity (Line 130, page 6)?

Response: First, all variants reported by MSK IMPACT undergo manual review at clinical sign out. Additionally, we confirm that manual review for all discordant variants was conducted for both MSK-IMPACT data and WGS data to include in silico genotyping for the mutations through mpileup, evaluation of coverage, strand bias and sequence quality metrics. Most importantly, we undertook experimental validation by repeating MSK-IMPACT sequencing on all samples with discordant mutations. Validation by MSK-IMPACT which uses the same assay and same analysis workflow would be subject to the same sequence and assay biases.

R5.Comment 11

Was erroneous alignment ruled out as a contribution to the difference between the HLA-A call between genome and targeted MSK-IMPACT sequencing (page 6, line 132)?

Response: We can confirm that the HLA-A L102Hfs*73 event was not due to erroneous alignment.

R5.Comment 12

12. Supplemental Figure 6 seems uninformative in that similarly described cancer histologies fail to cluster discretely in the supplied TSNE. Can the authors comment on why this is the case? Perhaps the categories are too broad with variability in the underlying driver alteration? It is also not clear what constitutes the in-house reference cohort.

Response: We thank the reviewer for the opportunity to clarify Supplementary Fig. 6. Contrary to the reviewer's point, the majority of tumors from the same cancer type do cluster together in Supplementary Fig. 6. Although we have few numbers of tumors from most cancer types, DSRCT, hepatoblastoma, Wilm's tumor, two CNS entities (pilocytic astrocytoma and GBM), ARMB tumors all cluster tightly together. Osteosarcomas and neuroblastomas, the two most frequent cancer types in our cohort exhibit higher variance in the t-SNE cluster where the majority of cases still cluster together but a subset of them separate (also together in distinct subclusters). We did not identify specific associations with driver mutations (e.g. MYC-N, TERT, ATRX), suggesting that this is likely attributed to transcriptional heterogeneity.

tSNE clustering includes 101 tumors from the cohort in the manuscript (dots with solid borders) and the inhouse reference consisting of 155 pediatric tumors not included in the current cohort (shown with dots with the same color without solid borders). We have now clarified this in the figure legend for Supplementary Fig. 6 as follows:

Supplementary Figure 6: RNA Clustering. a) t-distributed Stochastic Neighbor Embedding (TSNE) map of RNA expression data from study (n=101) and extended in-house reference cohort (n=155) colored by selected disease groups. b) Left bar plot shows breakdown of patients with ≥ 1 expression biomarker

supported by an SV event while right bar plot shows the different categories of associated SV events (Homo. Loss = Homozygous Loss).

and the methods section as follows:

tSNE was performed on RNA-seq data from 101 tumors from the cohort and an inhouse reference consisting of 155 pediatric tumors using python scikit-learn (v0.21.1, <https://scikit-learn.org/>) and visualized interactively using python bokeh (v1.2.0, <https://docs.bokeh.org/>)³⁶.

R5.Comment 13

References 1 and 2 are identical in the methods, as are references 5 and 8. Please check references throughout for accuracy.

Response: We thank the reviewer for identifying this error and providing us with the opportunity to fix it.

R5.Comment 14

S.Figure1 a should have a legend to describe the disease type displayed in the chart.

Response: This legend for this figure has been fixed now.

R5.Comment 15

Supplementary Table 1, convert age from days to years.

Response: The age has been converted from days to years in Supplementary Table 1.

Reviewer #5 Response to Reviewer #3

As requested by the editor, this portion of the review serves as an assessment to the degree to which the authors have responded to the comments from Reviewer 3.

R5 on R3C1

Reviewer 3. Comment 1: The overall impression of the manuscript is that it largely achieves the stated goal of supporting the clinical utility and feasibility of WGS for identifying somatic alterations and mutational signatures, but that it gives little to no attention to these areas for germline susceptibility or for the impact of the transcriptome.

Reviewer 5. R3C1: I am in agreement with reviewer 3, that the authors still substantially focus on somatic genome data, rather than explicitly emphasizing the value of RNASeq and germline components. As related to germline, the authors highlight extension of detection of genome sequencing in picking up 4 germline variants not detected by the MSK-IMPACT panel. Exome would certainly be anticipated to pick up these variants, therefore the lack of coverage due to the design of the MSK-IMPACT panel which is declared as standard of care by the authors is a moot point. The limited data presented herein do not extend the utility of genome for germline variation.

Response: A key message of our manuscript is the utility of RNA-seq in the detection, validation and prioritization of clinically relevant mutations. We address this in our response to R4 Comment 7.

However, the value of RNA-Seq findings in directing clinical decisions is less established and thus warrant further validation prior to consideration in clinical guidelines.

In response to R3 Comment 1 we implemented the method from PMID: 34112699, to enumerate the presence of gene expression biomarkers (Supplementary Table 10).

In this version of the manuscript, in response to R5 comments we perform for the first time a systematic integration of these putative expression biomarkers with SV findings from WGS. In Supplementary Table 3 we provide a breakdown of the expression biomarkers into those that 1) are tissue-specific 2) overlap into regions rearranged by structural variants or 3) neither. This analysis shows that in 29% of patients (29/101), at least 1 gene with tissue-specific expression was over-expressed in a tumor originating from that tissue and in 55/101 patients, at least 1 gene falling within a genomic region rearranged by structural variants had an aberrant expression. To this end, we have changed Figure-4, Supplementary Fig. 2, SSupplementary Fig. 4. We also updated the results section to include:

Global gene expression signatures were further used to cluster samples by tumor type, providing further opportunity to resolve a patient's diagnosis (Supplementary Fig. 6a). Last, in the 101 patients with RNA-seq data we identified on average 18 gene expression biomarkers per sample (range=1-99)¹⁶

(Supplementary Table 10). However, the evidence with regards to the clinical utility of such expression biomarkers remains to be validated. To this end, we performed a systematic interrogation of genes with aberrant expression with known tissue-specific expression patterns and SVs or fusions detected from cWGTS. Overall only 8% of the expression biomarkers were associated with an acquired SV or fusion gene, demarcating a subset of high confidence expression biomarkers (average per patient = 1, range= 0-10). This limits the number of patients in our cohort with an expression biomarker supported by an SV to 54% (n=55) (Supplementary Fig. 6b). Demonstrating the utility of this integrative analysis, we identified a concomitant KRAS amplification and overexpression in two patients with no clinically relevant biomarkers (H135462 and H195916) by clinical testing. Of note in both patients the amplification was not reported by MSK-IMPACT pointing to the lower sensitivity to detect copy number changes by panel-based assays.

We agree with the reviewer that the germline findings would have been picked up by exome and in fact we have accounted for these events in the % of events that would have been detected by exome sequencing (Figure 4a).

The first FDA-approved whole exome test for germline profiling (Helix platform) was not available at the time of study design. Furthermore, there are no standardized guidelines for germline work up of cancer patients. In contrast, panel testing (e.g Foundation medicine, Tempus, MSK-IMPACT) does represent the most common form of tumor profiling in clinical oncology and our comparator, MSK-IMPACT is the first NGS-based mutational profiling assay approved by the FDA in 2017 for both somatic and germline mutations and thus is representative of standard of care.

Importantly a key message of the manuscript is that cWGTS represents an assay that captures all clinically relevant mutations (germline and somatic) in a single integrative workflow and our data are in support with this statement. Last, exome would not be in a position to detect non-coding germline mutations (e.g TERT promoter, or germline SVs).

Reviewer 5. R3C1: Additionally, as related to the transcriptome, it is unclear how Supp. Figure 6 and Supp. Table 9 extend valuable information to the reader. Supplemental Figure 6 seems uninformative in that similarly described cancer histologies fail to cluster discretely in the supplied TSNE. Perhaps the categories are too broad with variability in the underlying driver alteration. It is also not clear what constitutes the in-house reference cohort. A UMAP may provide more meaningful information, with regard to validity in clustering of similar tumor types. Supp. Table 9 contains information of relevance, but this should all be contained in a master table which displays the tumor/patient identifier, clinically relevant germline and somatic SNVs/indels, oncogenic germline and somatic SNVs/Indels, fusions, and expression outliers for ease of a reader in being able to understand if altered expression is correlated with genomic findings, vs consistent with a given tissue/tumor type.

Response: We address the concern on TSNE clustering in our response to comment-12.

We also confirm that we have now updated Supplementary Table 3 with the fields requested by the reviewer.

R5 on R3C2

Reviewer 3. Comment 2: In fact, there seems to be some "drift" from the stated goals of the study that are emphasized in the abstract when compared to what is presented in the text, including a brief foray into comparing primary to recurrent cancers, as well as a very small number of samples evaluated through WGS of circulating free DNA when compared to the WGS analysis results of the tumor DNA. As such, it seems the manuscript needs to be more focused in order to effectively support the clinical utility and feasibility goals that are stated.

Reviewer 5. R3C2: The authors partially address comments as related to reviewer 3's concerns. I am in agreement that focus could improve this manuscript further. For example, at times the authors compare results of genome sequencing against that of a targeted panel test, which they consider to be standard of care. This is one institution's standard of care. It is of greater relevance to compare against an exome which is increasingly being utilized in a clinical setting. While comparison against an exome is performed for some portions, and should be the benchmark for which increased yield as related to the genome is standardized.

Response: As we stated in our response to the comment-1, in our study we compare the clinical utility of cWGTS as compared to standard of care clinical profiling assays. This is consistent with emerging studies evaluating clinical utility of WGS [PMID: 33964451]. To our knowledge there is no FDA approved WES-based assay for somatic mutation detection and the vast majority of tumor profiling in oncology is centered around panel tests. The comparator assay in our paper (MSK-IMPACT) represents the first FDA approved panel test and is one of the most comprehensive panel tests in clinical oncology. Furthermore, although WES may be state of the art for mutation calling in coding regions, FDA-authorization of multiple panel tests and reimbursement by payers underscores panel sequencing as standard of care.

Our choice of a targeted panel assay as a comparator is further justified because: 1. The published literature evaluating the clinical utility of targeted NGS panels and whole exome sequencing in pediatric cancers demonstrates largely similar findings without clear additional benefit of exome sequencing (PMID: 28007021, 26822237, 26822149); 2. Head-to-head comparison of WES and WGS analyses demonstrated that WES is not superior to WGS in calling coding mutations with a similar proportion of subclonal variants attributed to tumor heterogeneity identified in both WES and WGS [PMID: 32958763]; 3. WGS identified 50% more variation in exonic regions due GC-bias seen in WES [PMID: 32958763]; 4. A similar analysis in pediatric tumors shows that WES is not suited to capture structural variants that play an important role in pediatric tumor pathogenesis [PMID: 30262806].

Whilst the main outline of the manuscript is focused on cWGTS comparisons to panel tests, we include one section that compares cWGTS to WES and RNA-Seq as we appreciate that this is of interest to the field.

Reviewer 5. R3C2: Additionally, there is limited value in providing information in this manuscript as related to cell-free genome studies. It would seem better to have a separate short report on those findings given the very limited sample size and the high degree of discordance. This will clearly become a valuable and worthwhile cohort to report on with regard to cell-free studies, however presently the small n limits interpretation and makes this portion lacking clarity for the reader. Also, Supp Table 10 is missing a Diagnosis field in column E. Additionally, check on the ability to retain full dates as related to procedures (re:HIPAA) in columns C and D. If retaining this dataset, consider updating to a span of days.

Response: The focus of the manuscript is on the utility of WGS in clinical oncology. We show that for some patients, suitable primary biopsies are challenging to access or limited by tumor purity. The cfDNA section demonstrates for the first time the possibility to derive full coverage WGS data and comprehensive mutation profiling to include copy number, small mutations and SV's from cfDNA. This opens the opportunity for comprehensive genome profiling for patients with limited biopsies, with high cfDNA content which is frequently the case for patients with progressive metastatic disease in need for therapeutic options. Thus, the thematic link and novelty of the analysis should justify retention of the cfDNA section in the manuscript.

In response to the reviewer's specific comments we would like to clarify the following points:

1. Our study, to our knowledge, is the first to demonstrate proof of principle that full coverage tumor WGS analysis to include copy number analysis, calling of small mutations, SVs, as well as derivation of mutation signatures and putative phylogenetic structures is possible from cfDNA. We acknowledge that this is a proof of concept analysis in the main text and discussion.
2. As reviewed recently in PMID: 31563277 which evaluated 50 different studies from literature focusing on pediatric tumors, ctDNA profiling in pediatric patients have been so far confined the analysis of genome-wide CNA profiles (18 studies), targeted sequencing of point mutations/indels (17 studies) or patient-specific translocations/rearrangements with a priori knowledge (13 studies).

Thus, having demonstrated the clinical utility of cWGTS in detecting clinically relevant events, the aim of our cfDNA analysis was to evaluate if cfDNA presents an additional bioanalyte to deploy WGS for diagnostic purposes and in our pilot we evaluated the feasibility of tumor-naive mutation/SV calling genome-wide is feasible. We argue that the novelty of our analysis presents a significant advance in the field. We fully acknowledge that definitive follow up studies on the analytical validation of cfDNA are warranted that extend beyond the scope of this paper and include in the following sentences:

In cfDNA results section:

Recently cfDNA NGS profiling including WGS has been used to detect tumor-specific CNAs and fragmentation patterns in pediatric tumors⁵⁴⁻⁵⁶. However, the potential of high-depth WGS in tissue-naive identification of the genetic changes (SNVs, indels, SVs) in a patient's cancer genome from cfDNA has been largely unexplored.

and

These results provide the first proof-of-concept for the feasibility of deriving tumor agnostic comprehensive WGS profiling from a liquid biopsy.

In discussion:

To this end we provide proof-of-concept feasibility data demonstrating that comprehensive WGS profiling can also be leveraged in patients who have high cfDNA content in circulation at time of diagnosis or relapse. Our findings pave the way for future studies focused on analytical validation and optimizations of comprehensive tumor agnostic WGS profiling from cfDNA for diagnostic purposes.

We also change the text in the abstract as follows:

Last, we demonstrate the potential of cfDNA for tissue-agnostic WGS profiling.

We also confirm that we did the changes in Supplemental Table 11 suggested by the reviewer.

R5 on R3C4

Reviewer 3. Comment 4: Finally, the figures are entirely too multi-part and in some cases, are poorly described by the figure legends including several that do not even have a title listed for the figure.

Reviewer 5. R3C4: The authors have partially addressed reviewer 3's concerns on figures. In addition to the below:

- Define "small mut" in Figure 4b as that term is not used elsewhere in the paper.
- In Figure 5a define genomic reference used as related to breakpoints provided.
- Define "TRA" in Figure 5e as that term is not used elsewhere in the paper.
- In Figure 5c legend, is this meant to say chromosomes 6, 9 and 18, rather than 6, 8, 18? I am thinking that is the case based on the chromosomes highlighted relative to your breakpoints.
- Figure 6 legend and throughout the paper/tables, used proper HGVS nomenclature to describe splice variants.
- Figure 6d, recommend to define what comprises categorical definition of the 89064 indels and 568 rearrangements. For example, what is "deletion other" or "m-homology", or "t. Duplication"?
- Supplementary Fig. 1a should have a legend to describe the disease type displayed in the chart.
- Supplementary Fig. 7e, for chromothripsis, define SV vs disruptive vs fusion

Response: We thank the reviewer for pointing these out and have fixed figures/legends/tables accordingly.

R5 on R3C5

Reviewer 3. Comment 5: The Introduction is brief and basically ignores the specific aspects of the pediatric/AYA cancer genome that make DNA plus RNA profiling so critical: common fusion drivers, low mutational burden, methylation-driven gene expression profiles. Fine-tuning this a bit to the challenges might support the notion of broader profiling of DNA and RNA a bit better as a precursor to presenting the results.

Reviewer 5. R3C4: This is partially addressed. Recommend to extend on the history here. Because pediatric cancers have some fundamental differences to those in the adult setting, more broadly applied studies of exome and/or transcriptome sequencing have been initiated for some time (Parsons, Baylor, BASIC3; PipSeq Columbia; St. Jude, PMID: 30262806).

Response: We thank the reviewer for raising this point. We have now included these WES-based studies and methylation profiling studies in our introduction:

Recent studies showed the utility of whole-exome sequencing (WES) in identifying coding mutations in rare cancer genes^{6,7}. However, for patients with pediatric or rare cancers that have low mutation burden, distinctive methylation profiles^{8,9} and are primarily driven by SVs and fusion genes, panel/WES tests fail to identify a clinical biomarker in most cases^{1-4,7}.

R5 on R3C6

Reviewer 3. Comment 6: In the results section, the timeline to results aspect is presented briefly for the 114 cases, yielding a best case scenario of 9 days to results, as assayed (apparently) during a 4 week period when there were 4 cases per week being profiled. Ostensibly, this was done once all the challenges of implementing the different 'moving parts' of the workflow were addressed, and represents a best-case scenario, which is fine. However, it would be more informative to present TAT during the interval over which all 114 patients were studied, discussing the range of TAT that was experienced, how that improved over the interval of the study being reported and what innovations or developments most significantly impacted the TAT. In particular, since the ISABL pipeline/process has already been reported, it would help to understand how profound its impact was on the overall TAT. I would basically suggest replacing the non-informative Figure 1a with a more realistic representation of the changes to the pipeline over time and their impact on TAT.

Reviewer 5. R3C6: The response by the authors does not fully address the highly valid comments made by Reviewer 3. The presented TAT data are highly stylized through the use of only 4 cases processed per week across a 4 week period (n=16). I am in agreement that presenting TAT across the initiation of testing in this cohort would be very meaningful to the reader. If a retrospective analysis is not possible, are there additional recent data which can be added which support sustained TAT of this nature?

Additionally, if the goal is for a clinically relevant TAT, it would be meaningful to include the time period between the interdisciplinary meetings and final, fully interpreted result.

Response: We refer the reviewer to our response to reviewer 4 comment 4 also included here:

To address the reviewer's concern we now provide the range of our TAT as well as best reported TAT in Fig. 1b and in the main text as follows:

Time logs were audited starting at the time of surgical biopsy submission to report delivery for review by an interdisciplinary molecular tumor board for samples in our study with audit trails recorded through our biosciences platform (Supplementary Fig. 1e). End-to-end, this workflow was executed on average in 17 days during the developmental phase (range: 11-29, n=59 samples), reaching a fully optimized workflow with a final turn around of 9 days (Fig. 1b, n=16).

Additionally, we provide end to end TAT audit trails for 59 samples, excluding the 16 cases that represent our final optimized TAT. Please note that owing to a transition into our new version of our bioscience platform we only have TAT for these samples. To address the reviewers concern we further provide provide a detailed breakdown of the TAT and the respective components in S. Figure 1e

Last, with regards to time to fully interpret the result. The report generated represents the final output of the process that is discussed at the biweekly inter disciplinary tumor board with no extra time required for report interpretation. This is included in the current version of the methods sections as follows:

Putative findings of clinical relevance identified by WGS and RNA-seq were reviewed by an interdisciplinary team of clinical oncologists, molecular pathologists and cancer genomics experts. Typically 8-10 cases were reviewed in an hour-long tumor board meeting that was held biweekly.

R5 on R3C9

Reviewer 3. Comment 9: The description of cWGTS that follows, again seems to focus on the somatic genome and provides only minimal information on how the RNA-seq is being evaluated; fusions and gene expression "signatures" (although these really aren't presented, only single gene overexpression is shown).

Reviewer 5. R3C9: Comment 5 from reviewer 3 surrounds a different topic than that raised below. I find this more closely associated to comment 1 for which I have provided feedback above.

Response: We refer the reviewer to our response to their own comment on Reviewer 3's comment-1 (R5 on R3C1).

R5 on R3C13

Reviewer 3. Comment 13: The labeling of figure 2g doesn't match the text description on page 7 lines 141-147 in terms of the tests used, so this is confusing. In figure 2h, while it is true that there are "novel findings" only possible from cWGTS, (really from WGS), over 50% of the novel findings are SVs, but what proportion of those were clinically meaningful vs. not?

Reviewer 5. R3C13: This has been partially addressed. While cWGTS analyses extended the detectable variant and signature spectrum, the description of the extension of findings is challenging to follow and difficult to discern from Figure 4a and Supp Table 2. This is particularly true when referencing that "less than half of the findings in cWGTS could be captured by WES and RNA-seq" (Line 176, page 8). Recommend to add to the text that the majority represented SV, and to clearly indicate in a column in Supp Table 2 which findings were detected by panel sequencing and which would have been captured by exome and RNA sequencing. Also, it is of interest for the reader to know if other well-described and clinically relevant genomic alterations were present in these exact samples which would have been captured by exome and RNA seq or if this new finding represented the only disease-associated finding of relevance. This speaks to understanding how broadly use of cWGTS extends the analysis for clinically relevant findings, vs adds to an existing broad repertoire of detected variation in a given sample using existing well-described techniques.

Response: We refer the reviewer to our response to their own comment-2.

R5 on R3C14

Reviewer 3. Comment 14: Fusions: were the novel fusions on page 9 validated? are they in-frame fusions?

Reviewer 5. R3C14: This has been partially addressed. Greater clarity for the reader could be provided in adding the tumor type to figure S. Figure 5. Additionally, a novel fusion does not necessarily constitute a driver event or an alteration of clinical relevance, even if in-frame and validated. Recommend to include an additional column in S. Figure 5 if either partner has been previously described in an in-frame fusion event within cancer, and within the specific tumor type. This is important given the author's point that "this speaks to the importance of WGTS vs WGS or RNA-seq in the clinical setting."

Response: We have updated Supplementary Table 6 with literature support for the fusion or fusion partner and added tumor type to Supplementary Fig. 6.

R5 on R3C21

Reviewer 3. Comment 21: Similarly, the study of cfDNA vs. WGTS (really WGS) of the tumor is interesting but the numbers of patients evaluated in this way are quite small and probably inconclusive.

Reviewer 5. R3C21: I am in agreement with Reviewer 3 regarding the paucity of data in support of cell-free DNA studies for inclusion in this manuscript. There is particular concern surrounding the lack of reproducibility of findings, which may be attributed to low cf fraction or other technical challenges in the assay. Recommend to remove.

Response: We refer the reviewer to our response to their own comment on reviewer 3's comment-2.

R5 on R3C22

Reviewer 3. Comment 22: Finally, it would be interesting to understand the proportion of patients for which additional, medically meaningful findings were obtained, either in the primary or recurrent setting, that indicated new therapeutic aspects not uncovered by prior testing. This is only one aspect of clinical benefit that would meaningfully support the advocacy for cWGTS.

Reviewer 5. R3C22: This has been partially addressed. It is unclear to what extent cWGTS actually extends clinical utility for diagnosis, prognosis, or therapeutic options. It is of interest for the reader to know if other clinically relevant genomic alterations were present which would have been captured by exome or RNA seq or if this a finding derived from cWGTS represented the sole disease-associated finding of relevance. This speaks to understanding how broadly use of cWGTS extends the analysis for clinically relevant findings, vs adds to an existing broad repertoire of detected variation in a given sample using existing well-described techniques. Supplementary Table 2 provides presence or absence of an identified event, however much greater interpretation and extension of the clinical utility of these data would occur if all oncogenic and clinically relevant variants are documented in the table, along with designation as to which were detected solely by cWGTS.

Response: We have modified Supplementary Table 3 (previously Supplementary Table 2) to include information on 1) if the additional finding can only be identified by cWGTS or WES+RNA-seq is sufficient 2) Clinically relevant small mutations, CNAs and SVs from MSK-IMPACT.

REVIEWERS' COMMENTS

Reviewer #4 (Remarks to the Author):

The authors have done a good job improving the manuscript, but only partially addressed the concerns.

Most importantly, the strong recommendation by multiple reviewers to remove the preliminary cfDNA experiments, which do not add to the main message of the paper nor are sufficiently powered and properly discussed regarding caveats and conditions, is fully ignored. Therefore, the claim regarding cfDNA in the abstract is scientifically insufficiently supported and in my view misleading.

Details of data availability are still lacking. Is data available for re-use and is it access-controlled? In what format is data made available and how does the access procedure work? dbGAP and cBioportal are mentioned in the manuscript, but are not platforms for making raw WGS and MSK-IMPACT data available, which would be required to reproduce findings.

Reviewer #5 (Remarks to the Author):

The authors have addressed and responded to reviewer comments raised in the prior reviews. Very minor omissions/edits are requested as listed below:

Supplemental Figure 4, C is missing a description of the color coding scheme in the legend.

For supplemental tables, excel tab labels for tables 9 and 10 are reversed relative to table descriptions.

The newly added Supplementary Data Table 2, in response to R5.Comment 4, provides clear and relevant metrics on sequencing performance for which I thank the authors. To improve clarity for the reader, could the authors please define "DNA GC Dropout" in column J.

With regard to Reviewer 5. R3C4: -Define "small mut" in Figure 4b as that term is not used elsewhere in the paper.

I do not see that this has been defined yet in the revision.

REVIEWERS' COMMENTS

Reviewer #4 (Remarks to the Author):

The authors have done a good job improving the manuscript, but only partially addressed the concerns.

Most importantly, the strong recommendation by multiple reviewers to remove the preliminary cfDNA experiments, which do not add to the main message of the paper nor are sufficiently powered and properly discussed regarding caveats and conditions, is fully ignored. Therefore, the claim regarding cfDNA in the abstract is scientifically insufficiently supported and in my view misleading.

Response:

We thank the reviewer for their comments. As per their suggestion

1. We removed the cfDNA from the abstract.
2. We modified the last paragraph of the cfDNA section to read

“We further demonstrate the potential of cfDNA to capture a comprehensive representation of different types of variants across the tumor phylogeny”

3. We have already stated in the discussion that these analyses were done in patients high ctDNA in circulation.

“To this end we provide proof-of-concept feasibility data demonstrating that comprehensive WGS profiling can also be leveraged in patients who have high cfDNA content in circulation at time of diagnosis or relapse. Our findings pave the way for future studies focused on analytical validation and optimizations of comprehensive tumor agnostic WGS profiling from cfDNA for diagnostic purposes. “

Details of data availability are still lacking. Is data available for re-use and is it access-controlled? In what format is data made available and how does the access procedure work?

dbGAP and cbiportal are mentioned in the manuscript, but are not platforms for making raw WGS and MSK-impact data available, which would be required to reproduce findings.

All the data required to reproduce these analyses are now accessible. Raw data for WGS and RNA-seq is available under restricted access at http://www.ncbi.nlm.nih.gov/projects/gap/cgi-bin/study.cgi?study_id=phs002620.v1.p1 and processed data for MSK-IMPACT are available at https://www.cbiportal.org/study/summary?id=mixed_kunga_msk_2022

Reviewer #5 (Remarks to the Author):

The authors have addressed and responded to reviewer comments raised in the prior reviews. Very minor omissions/edits are requested as listed below:

Supplemental Figure 4, C is missing a description of the color coding scheme in the legend.

We have modified this panel to include a legend.

For supplemental tables, excel tab labels for tables 9 and 10 are reversed relative to table descriptions.

We have fixed the labels for these source data tables.

The newly added Supplementary Data Table 2, in response to R5.Comment 4, provides clear and relevant metrics on sequencing performance for which I thank the authors. To improve clarity for the reader, could the authors please define "DNA GC Dropout" in column J.

We have included a legend for all source data files in the supplementary information document. The indicated column is described in the legend of Source Data 2 now.

With regard to Reviewer 5. R3C4: -Define "small mut" in Figure 4b as that term is not used elsewhere in the paper.

I do not see that this has been defined yet in the revision.

We have given descriptions of this term in Fig-4b legend as well as all abbreviations we used in the figures.